# Cardiac fibroblasts regulate the development of heart failure via Htra3-TGF-β-IGFBP7 axis

Toshiyuki Ko[1,2,9], Seitaro Nomura [1,2,9 ✉], Shintaro Yamada[1,9], Kanna Fujita[1,2], Takanori Fujita[2], Masahiro Satoh[2,3], Chio Oka [4], Manami Katoh[2], Masamichi Ito[1], Mikako Katagiri[1], Tatsuro Sassa[1], Bo Zhang[1], Satoshi Hatsuse[1], Takanobu Yamada[1], Mutsuo Harada[1], Haruhiro Toko[1], Eisuke Amiya[1], Masaru Hatano[1], Osamu Kinoshita[5], Kan Nawata[6], Hiroyuki Abe [7], Tetsuo Ushiku[7], Minoru Ono[5], Masashi Ikeuchi[8], Hiroyuki Morita[1], Hiroyuki Aburatani [2 ✉] & Issei Komuro [1 ✉]

Tissue fibrosis and organ dysfunction are hallmarks of age-related diseases including heart failure, but it remains elusive whether there is a common pathway to induce both events. Through single-cell RNA-seq, spatial transcriptomics, and genetic perturbation, we elucidate that high-temperature requirement A serine peptidase 3 (Htra3) is a critical regulator of cardiac fibrosis and heart failure by maintaining the identity of quiescent cardiac fibroblasts through degrading transforming growth factor-β (TGF-β). Pressure overload downregulates expression of Htra3 in cardiac fibroblasts and activated TGF-β signaling, which induces not only cardiac fibrosis but also heart failure through DNA damage accumulation and secretory phenotype induction in failing cardiomyocytes. Overexpression of Htra3 in the heart inhibits TGF-β signaling and ameliorates cardiac dysfunction after pressure overload. Htra3-regulated induction of spatio-temporal cardiac fibrosis and cardiomyocyte secretory phenotype are observed specifically in infarct regions after myocardial infarction. Integrative analyses of single-cardiomyocyte transcriptome and plasma proteome in human reveal that IGFBP7, which is a cytokine downstream of TGF-β and secreted from failing cardiomyocytes, is the most predictable marker of advanced heart failure. These findings highlight the roles of cardiac fibroblasts in regulating cardiomyocyte homeostasis and cardiac fibrosis through the Htra3-TGF-β-IGFBP7 pathway, which would be a therapeutic target for heart failure.

[1] Department of Cardiovascular Medicine, Graduate School of Medicine, The University of Tokyo, Tokyo, Japan. [2] Genome Science Division, Research Center for Advanced Science and Technology, The University of Tokyo, Tokyo, Japan. [3] Cardiovascular Division, Brigham and Women's Hospital, Harvard Medical School, Boston, Massachusetts, USA. [4] Laboratory of Functional Genomics and Medicine, Nara Institute of Science and Technology, Nara, Japan. [5] Department of Cardiac Surgery, Graduate School of Medicine, The University of Tokyo, Tokyo, Japan. [6] Department of Cardiovascular Surgery, St. Marianna University School of Medicine, Kawasaki, Kanagawa, Japan. [7] Department of Pathology, Graduate School of Medicine, The University of Tokyo, Tokyo, Japan. [8] Department of Biodesign, Institute of Biomaterials and Bioengineering, Tokyo Medical and Dental University, Tokyo, Japan. [9] These authors contributed equally: Toshiyuki Ko, Seitaro Nomura, Shintaro Yamada. ✉email: senomura-cib@umin.ac.jp; haburata-tky@umin.ac.jp; komuro-tky@umin.ac.jp

Tissue fibrosis and organ dysfunction are hallmarks of age-related diseases including heart failure[1–5]. Pathological stresses on the heart induce fibroblast activation and collagen secretion[6], leading to cardiac fibrosis[7,8]. By contrast, repression of fibroblast activation can alleviate cardiac fibrosis and ameliorate the cardiac function after injury[9], suggesting the pathogenicity of fibrosis in heart failure. Persistent pathological stresses on the heart also induce accumulation of DNA damage and activation of p53 signaling in cardiomyocytes, leading to a phenotypic conversion to failing cardiomyocytes[10–14], which are characterized by mitochondrial dysfunction[15], autophagic dysfunction[16], elongated structure[12], and telomere shortening[17]. Inhibition of p53 or senolysis can improve the cardiac phenotypes after injury or with aging[12,18], suggesting that senescence-like failing cardiomyocytes cause heart failure. Thus, tissue fibrosis and cellular senescence, hallmarks of many age-related diseases, cause the development and progression of heart failure, but it remains elusive whether there is a common pathway to induce both events.

In this work, we hypothesized that cell-cell communications between cardiac fibroblasts and cardiomyocytes might be important to induce cardiac fibrosis and heart failure[19,20]. A cell-cell communication map of the heart using single-cell RNA-seq showed that cardiac fibroblasts have strong interactions with various types of cardiac cells including cardiomyocytes. Through single-cell co-expression network analysis, we identified high-temperature requirement A serine peptidase 3 (Htra3) in cardiac fibroblasts as a critical regulator of cardiomyocyte homeostasis as well as cardiac fibrosis.

## Results

**Single-cell network analysis identifies Htra3 as a central molecule of cardiac fibroblasts.** To investigate molecular interactions leading to the induction of failing cardiomyocytes, we isolated cardiomyocytes and non-cardiomyocytes from mice at 2 weeks after transverse aortic constriction (TAC) or sham surgery and performed single-cell RNA sequencing (scRNA-seq) with droplet-based encapsulation using the 10X Genomics platform and in-depth transcriptional phenotyping by Smart-seq2 profiling[21] (Fig. 1a and Supplementary Fig. 1a–d). Using 1,019 expression profiles of ligands and receptors (LR) from all datasets, we generated a co-expression network map and revealed the cell-type–specific transcriptional properties of LR in the heart (Fig. 1b). We next used an LR interaction database[22] to reconstruct the LR interaction network map in the heart and showed the potential comprehensive interactions between the same or different cell types (Fig. 1c, Table S1 and Supplementary Data 1). Pathway analysis revealed that specific signaling pathways such as the PI3K-Akt, Rap1 and TGF-β signaling pathways were robustly activated in the heart (Fig. 1d). Among the various cell types, cardiac fibroblasts most strongly interacted with multiple cell types, including cardiomyocytes (Fig. 1e). Pressure overload by TAC increased the cardiac fibroblast population, in which extracellular matrix-related genes such as *Ctgf, Tgfb1*, and *Fbn1* were activated (Fig. 1f and Supplementary Fig. 1e).

To identify significant gene modules characteristic for each cell type, we conducted single-cell co-expression network analysis[12,23] (Supplementary Fig. 1f–h). By generating a cardiac fibroblast network, we identified *high-temperature requirement A serine peptidase 3* (*Htra3*) as a molecule located at the center of the network (Fig. 1g, Supplementary Fig. 1h and Supplementary Data 2), together with TGF-β signaling-associated molecules such as *Dcn* (decorin) and *Islr* (Meflin)[24,25]. *Htra3* expression was strongly correlated with expression of the cardiac fibroblast module (Fig. 1h), suggesting that Htra3 defines the identity of

cardiac fibroblasts. Although it has been reported that *Htra3* is specifically expressed in the placenta and heart and is involved in placental development[26,27] (Supplementary Fig. 1j), its role in the heart is not known. Cardiac fibroblast-specific *Htra3* expression was confirmed by scRNA-seq (Fig. 1i and Supplementary Fig. 1b) and by RNA in situ hybridization with immunostaining of Pdgfr-α, which is a well-known cardiac fibroblast marker protein (Fig. 1j). Western blot analysis also showed that Htra3 protein was detected specifically in non-cardiomyocytes including cardiac fibroblasts (Supplementary Fig. 1k). Besides, we also confirmed *HTRA3* expression in the human cardiac interstitium (Fig. 1k).

**Cardiac fibroblast Htra3 governs cardiac homeostasis and is downregulated by pressure overload.** To investigate the functional role of Htra3 in the heart, we generated *Htra3* knockout (KO) mice (Supplementary Fig. S2a). Compared with wild-type (WT) control mice, *Htra3* KO mice showed cardiac hypertrophy with enlarged cardiomyocyte size even in the absence of pressure overload without change of blood pressure (Fig. 2a–c, Supplementary Fig. 2b, c). Additional mild pressure overload by TAC rapidly induced severe heart failure in *Htra3* KO mice but not in WT mice (Fig. 2b). There were no obvious morphological and histological differences between WT and *Htra3* KO mice in major organs other than the heart (Supplementary Fig. 2d). Sirius Red/ Fast Green collagen staining showed that TAC induced more severe cardiac fibrosis in *Htra3* KO mice than in WT mice (Fig. 2d). Pressure overload reduced the expression levels of *Htra3* in cardiac fibroblasts in WT mice (Fig. 2e). scRNA-seq of the human heart confirmed the specific expression of *HTRA3* in cardiac fibroblasts and its reduced expression in the failing heart[4] (Fig. 2f, Supplementary Fig. 2e, f). Mechanical stretch on isolated primary cardiac fibroblasts significantly repressed expression of *Htra3* and increased expression of *Tgfb1* (Fig. 2g). These results suggest that Htra3, specifically expressed in cardiac fibroblasts, is essential for maintaining cardiac size at basal state and functions against hemodynamic overload and that mechanical stress on cardiac fibroblasts decreases expression of *Htra3*.

**Htra3-induced TGF-β degradation is essential for the prevention of heart failure and fibrosis.** To elucidate the roles of Htra3 in the phenotype of cardiac fibroblasts, we used WT and *Htra3* KO mice at 2 weeks after TAC or sham surgery and performed full-length scRNA-seq (Smart-seq2) of cardiac fibroblasts isolated using an antibody against Pdgfr-α (Fig. 1a and Supplementary Fig. 3a). Uniform Manifold Approximation and Projection (UMAP) plotting and graph-based clustering separated cardiac fibroblasts into three clusters (C1-3) and pseudotime analysis revealed two trajectories (Fig. 3a, b). Fibroblast trajectory 1 (from C1 to C2) was induced mainly by *Htra3* deletion, and fibroblast trajectory 2 (from C1 to C3) was induced by pressure overload with or without *Htra3* deletion (Fig. 3a–c). Co-expression network analysis and random forest analysis clarified that modules M1 and M17, which were significantly involved in cell classification (Supplementary Fig. 3b, c), were mutually exclusive in their module activity (Fig. 3d).

Module M1 contained genes involved in innate immunity and the Toll-like receptor signaling pathway, including *Tlr2* and *Tlr4* (Fig. 3e, f), which are shown to be expressed in cardiac fibroblasts[28]. The activity of module M1 was strongly suppressed during trajectories 1 and 2 (Fig. 3e–g and Supplementary Fig. 3d), both of which were associated with *Htra3* downregulation, suggesting that Htra3 is critical for maintaining the state of quiescent cardiac fibroblasts[6,29].

Module M17 was enriched with genes involved in extracellular matrix formation and the TGF-β signaling pathway, and its

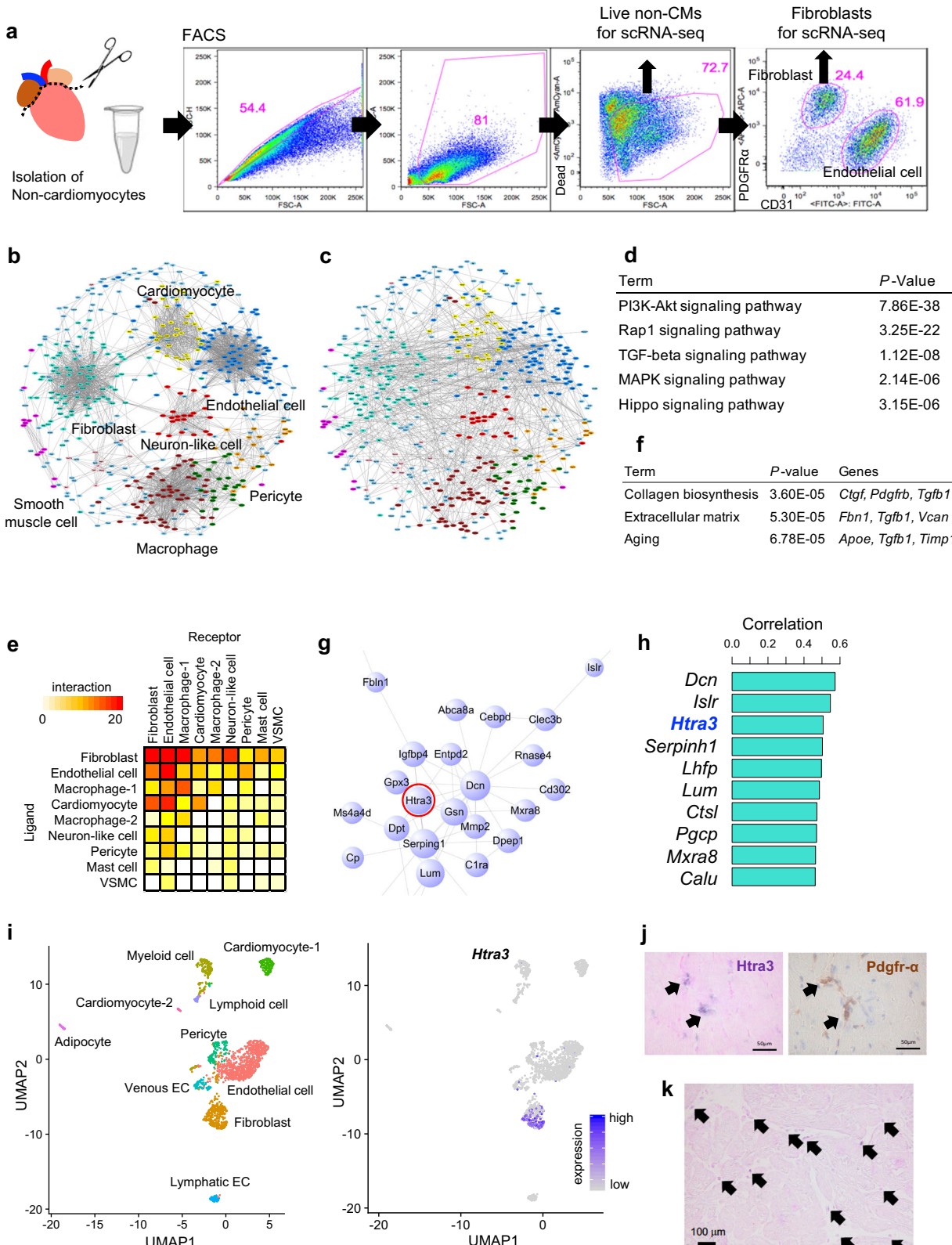

activity was upregulated in both trajectories 1 and 2 with different degrees (Fig. 3h–j and Supplementary Fig. 3d). Either pressure overload or *Htra3* deletion activated M17 and together they synergistically enhanced its activity (Fig. 3j). Expression of collagen fibril organization genes, such as *Col1a1* and *Col3a1*, was activated in the both trajectories, whereas expression of TGF-β signaling molecules, such as *Tgfb3* and *Tgfbr2*, and *Postn*, a marker of activated cardiac fibroblasts[6], was activated only in trajectory 2 (Fig. 3i). These findings indicate that Htra3 basically inhibits TGF-β signaling and that pressure overload and *Htra3* deletion synergistically activates TGF-β signaling, leading to activation of fibroblasts.

**Fig. 1 Single-cell network analysis identifies cardiac fibroblast Htra3. a** Isolation of non-cardiomyocytes and fibroblasts for scRNA-seq using fluorescence-activated cell sorting (FACS). Sorted live non-cardiomyocytes (non-CMs) were used for scRNA-seq using the 10X Genomics platform. Sorted fibroblasts with an antibody against PDGFR-α were used for scRNA-seq using the full-length Smart-seq2. **b, c** Ligand-receptor interaction map in the heart. Genes of ligands and receptors were projected on a two-dimensional map based on their correlation structure. Cell types were annotated by cell-type-specific expression profiles of genes assigned to each module. Edges indicate correlation (**b**) or interaction (**c**). **d** Pathway analysis of genes illustrated on the interaction map in the heart. *p*-values are determined by Fisher's Exact test. **e** Heatmap showing the number of interactions between cell types in the heart. VSMC, vascular smooth muscle cells. **f** Gene ontology (GO) analysis of upregulated genes after pressure overload in cardiac fibroblasts. Representative genes are also shown. *p*-values are determined by Fisher's Exact test. **g** Co-expression network of cardiac fibroblasts. **h** Bar graph showing the top 10 genes most correlated with the cardiac fibroblast module. **i** Uniform Manifold Approximation and Projection (UMAP) plot of single-cell transcriptomes of non-cardiomyocytes ($n = 1783$) from the murine heart ($n = 2$). Cell-type annotation (left) and *Htra3* expression (right) are on the UMAP plot. **j** RNA in situ hybridization of *Htra3* and immunostaining of Pdgfr-α using the paired mirror cardiac sections. Arrows indicate the colocalization of the *Htra3* and Pdgfr-α in the same cells. **k** RNA in situ hybridization of *HTRA3* in the human heart. Arrows indicate HTRA3 expression.

We next examined how Htra3 inhibits TGF-β signaling in vitro and in vivo. Biochemical analysis using primary cardiac fibroblasts revealed that *Htra3* overexpression reduced protein levels of both TGF-β1 and phospho-Smad2/3, a marker of activated TGF-β signaling (Fig. 3k, Supplementary Fig. 3e). Htra3 also directly bound to TGF-β (Fig. 3l), suggesting that Htra3 inhibits TGF-β signaling by degrading TGF-β. Phospho-Smad2/3 immunostaining of heart tissue showed that either *Htra3* deletion or TAC activated Smad2/3, and their combination synergistically activated Smad2/3 not only in cardiac fibroblasts but also in cardiomyocytes (Fig. 3m). Furthermore, TGF-β–neutralizing antibody remarkably ameliorated systolic dysfunction as well as cardiac fibrosis of *Htra3* KO mice after TAC surgery (Supplementary Fig. 3f, g). Besides, TGF-β–neutralizing antibody injection from 4 weeks old also ameliorated ventricular hypertrophy (Supplementary Fig. 3h). These results suggest that Htra3-induced TGF-β degradation is essential for the prevention of heart failure and fibrosis.

**Htra3 repression-induced activation of TGF-β signaling promotes the induction of senescent failing cardiomyocytes.** To understand how activation of TGF-β signaling affects cardiomyocytes and leads to heart failure, we performed scRNA-seq of cardiomyocytes isolated from WT and *Htra3* KO mice at 2 weeks after TAC or sham surgery. UMAP plotting classified cardiomyocytes into 4 clusters, and pseudotime analysis revealed 2 trajectories (Fig. 4a, b, Supplementary Fig. 4a). Co-expression network analysis and random forest analysis identified modules M1 and M2 as significantly involved in cell classification (Supplementary Fig. 4b, c). *Htra3* deletion, TAC surgery, and their combination induced the transition of cardiomyocytes from C1 to C2, C3, and C4, respectively (Fig. 4c). Cardiomyocyte trajectories 1 and 2 correspond to the transitions from C1 to C3 via C2 and from C1 to C4 via C2, respectively (Fig. 4b).

Module M1, enriched with genes involved in mitochondrial oxidative phosphorylation and DNA repair (e.g., *Atm*, *Parp1/2*, and *Rpa2*) (Fig. 4d), was significantly suppressed during both trajectories 1 and 2 (Fig. 4d–f, Supplementary Fig. 4d), suggesting that activation of TGF-β signaling suppresses the expression of genes related to mitochondria and DNA repair (Fig. 4d, e). In cardiomyocytes from *Htra3* KO mice, genes involved in oxidative-reduction process were down-regulated, and genes involved in protein synthesis were up-regulated without pressure overload (Supplementary Fig. 4e), which may lead to cardiomyocyte hypertrophy in *Htra3* KO mice (Fig. 2c).

Module M2, which contains genes related to small GTPase signaling (e.g., *Rhob* and *Rac1*), TGF-β receptor signaling (e.g., *Ltbp4* and *Tgfbr2*), and p53 signaling (e.g., *Trp53* and *Cdkn1a*), was specifically activated in C4 through trajectory 2 (Fig. 4g–i). A combination of *Htra3* deletion and pressure overload significantly promoted the induction of failing cardiomyocytes stained with

antibodies against γH2A.X and p21[12] (Fig. 4j, Supplementary Fig. 4f). Immunostaining also showed increased DNA damage accumulation in cardiac fibroblasts of *Htra3* KO mice after TAC surgery (Supplementary Fig. 4g, h), suggesting that Htra3 repression-induced activation of TGF-β signaling increased DNA damage not only in cardiomyocytes but also in cardiac fibroblasts. We also confirmed the increased expression of DNA damage-related genes in cardiomyocytes of *Htra3* KO mice after TAC operation (Fig. 4k). M2 also contained various kinds of secretory factors, including *Bmp1*, *Cxcl12*, and *Igfbp7*, suggesting that cardiomyocytes expressing M2 show DNA damage-induced secretory phenotype, similar to senescence-associated secretory phenotypes (SASP) (Fig. 4l). Clustering analysis identified that module M9, enriched with genes that encode proteins related to collagen fibril organization (e.g., *Col1a1*, *Col3a1*, and *Fbn1*) and TGF-β signaling (e.g., *Tgfb3*, *Nox4*, and *Postn*), was also specifically activated in C4 like M2 (Fig. 4m–o, Supplementary Fig. 4i).

**TGF-β-induced Nox4 expression and subsequent p53 activation are essential for the induction of failing cardiomyocytes with secretory phenotype.** To elucidate whether TGF-β-induced activation of *Nox4*[30] in M9, a principal ROS-generating NADPH oxidase[31,32] in cardiomyocytes[33], is involved in increased DNA damage and the development of heart failure, we injected *Nox4*-shRNA adeno-associated virus 9 (AAV9) vector to suppress *Nox4* expression (Fig. 5a). Echocardiography revealed that knockdown of *Nox4* rescued cardiac dysfunction in *Htra3* KO mice after TAC surgery (Fig. 5b, Supplementary Fig. 5a). Immunostaining and western blot analysis showed that Nox4 suppression significantly reduced DNA damage (Fig. 5c, d) and expression of TGF-β signaling-related molecules (Fig. 5e). Nox4 overexpression in the heart using the AAV9 vector increased the number of γH2A.X positive cardiomyocytes and significantly exacerbated cardiac dilatation and function (Supplementary Fig. 5b, c), consistent with the previous report[33]. scRNA-seq of cardiomyocytes isolated from cardiomyocyte-specific p53 KO mice after TAC surgery[12] showed that p53 deletion suppressed TGF-β signaling-related molecules (e.g., *Tgfb3* and *Tgfbr2*) and secretory factors (e.g., *Cxcl12* and *Igfbp7*) in M2 (Supplementary Fig. 5d). We next injected AAV9-*Htra3* vector into WT mice at 1 week after TAC surgery. *Htra3* overexpression in the heart inhibited TGF-β signaling and ameliorated cardiac dysfunction as well as fibrosis (Fig. 5f–j). These results suggest that TGF-β signaling induces accumulation of DNA damage and subsequent activation of p53 signaling by repressing DNA repair genes and promoting Nox4-mediated ROS production, resulting in the induction of senescent failing cardiomyocytes, which are characterized by the expression of TGF-β signaling-related molecules and secretory factors. Htra3-induced TGF-β repression might be a therapeutic approach for heart failure.

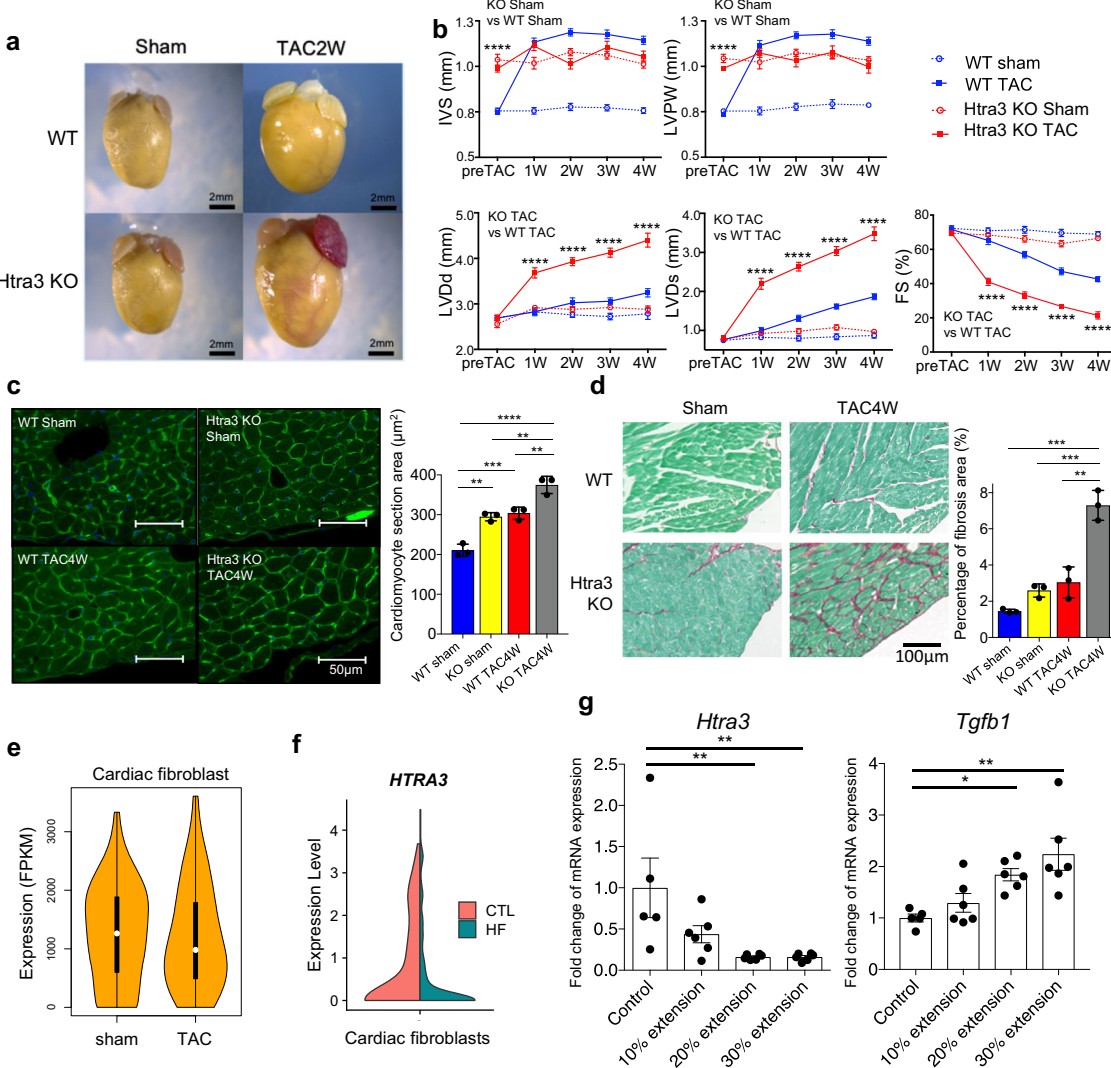

**Fig. 2 *Htra3* KO mice shows cardiac and cardiomyocyte hypertrophy and are vulnerable to mechanical stress. a** Representative cardiac morphology of WT and *Htra3* KO mice at 2 weeks after TAC or sham surgery. **b** Echocardiographic assessment of the heart of WT and Htra3 KO mice after TAC or sham surgery. $n = 11$ (WT Sham), $n = 15$ (WT TAC), $n = 10$ (Htra3 KO Sham), $n = 15$ (Htra3 KO TAC). Data are shown as mean and SD. *$P < 0.05$, **$P < 0.01$, ***$P < 0.005$, ****$P < 0.001$; significance was determined by two-way analysis of variance (ANOVA) with Bonferroni's multiple comparison test. Source data are provided as a Source Data file. **c**, Assessment of cardiomyocyte section area. WGA and DAPI are used to stain the plasma membrane and nucleus, respectively. Results of quantitative analysis are also shown. Averaged data from about 40–100 cells per heart ($n = 3$ each). Data are shown as mean and SD. **$P < 0.01$, ***$P < 0.005$, ****$P < 0.001$; significance was determined by one-way ANOVA with Tukey's or Dunnett's post hoc test. Source data are provided as a Source Data file. **d** Histochemical detection of collagen fibers by Sirius Red/Fast Green dye staining in WT and Htra3 KO mice after TAC or sham surgery. Results of quantitative analysis of fibrosis area are also shown ($n = 3$ each). Data are shown as mean and SD. *$P < 0.05$, **$P < 0.01$, ***$P < 0.005$, ****$P < 0.001$; significance was determined by two-way ANOVA with Tukey's or Dunnett's post hoc test. Source data are provided as a Source Data file. **e** Violin plot showing Htra3 RNA expression in cardiac fibroblasts at 2 weeks after TAC ($n = 40$) and sham surgery ($n = 109$) in WT mice. Data represent box plots and individual data points. Box plots show the median (center line), first and third quartiles (box edges), while the whiskers going from each quartile to the minimum or maximum. **f** Violin plot showing the *HTRA3* expression levels ($n = 143$ cells from control subjects (CTL), $n = 151$ cells for patients with heart failure (HF)). **g** mRNA expression levels of Htra3 and Tgfb1 in cultured cardiac fibroblasts after the 0~30% mechanical extension were assessed by real-time qPCR ($n = 5, 6, 6, 6$ at each group, respectively). Data are shown as mean and SD. *$P < 0.05$, **$P < 0.01$; Significance was determined by one-way analysis of variance (ANOVA) with Bonferroni's multiple comparison test. Source data are provided as a Source Data file.

**Spatial transcriptome reveals that the Htra3-TGF-β axis is essential for the prevention of cardiac fibrosis and cardiomyocyte secretory phenotype induction in infarct regions after myocardial infarction**. To elucidate how Htra3 spatially regulates cardiac remodeling, we investigated the role of Htra3 in myocardial infarction (MI) model. Spatial transcriptomic analysis using Visium (10X Genomics) showed high expression of Htra3 in the infarct zone after MI, suggesting the possibility of spatial regulation in cardiac remodeling (Fig. 6a). *Htra3* KO mice

showed severe cardiac remodeling after MI compared with WT mice (Fig. 6b). Echocardiographic analysis revealed that *Htra3* KO mice showed significant cardiac dilatation as well as contractile dysfunction (Fig. 6c). Spatial transcriptomic analysis on cardiac tissues from WT and *Htra3* KO mice after sham or MI operation identified specific gene clusters which were characterized by spatial localization (Fig. 6d) and specific expression profiles (Fig. 6e, Supplementary Fig. 6a). MI induced expression of cluster 1 genes, which were enriched with genes expressed in

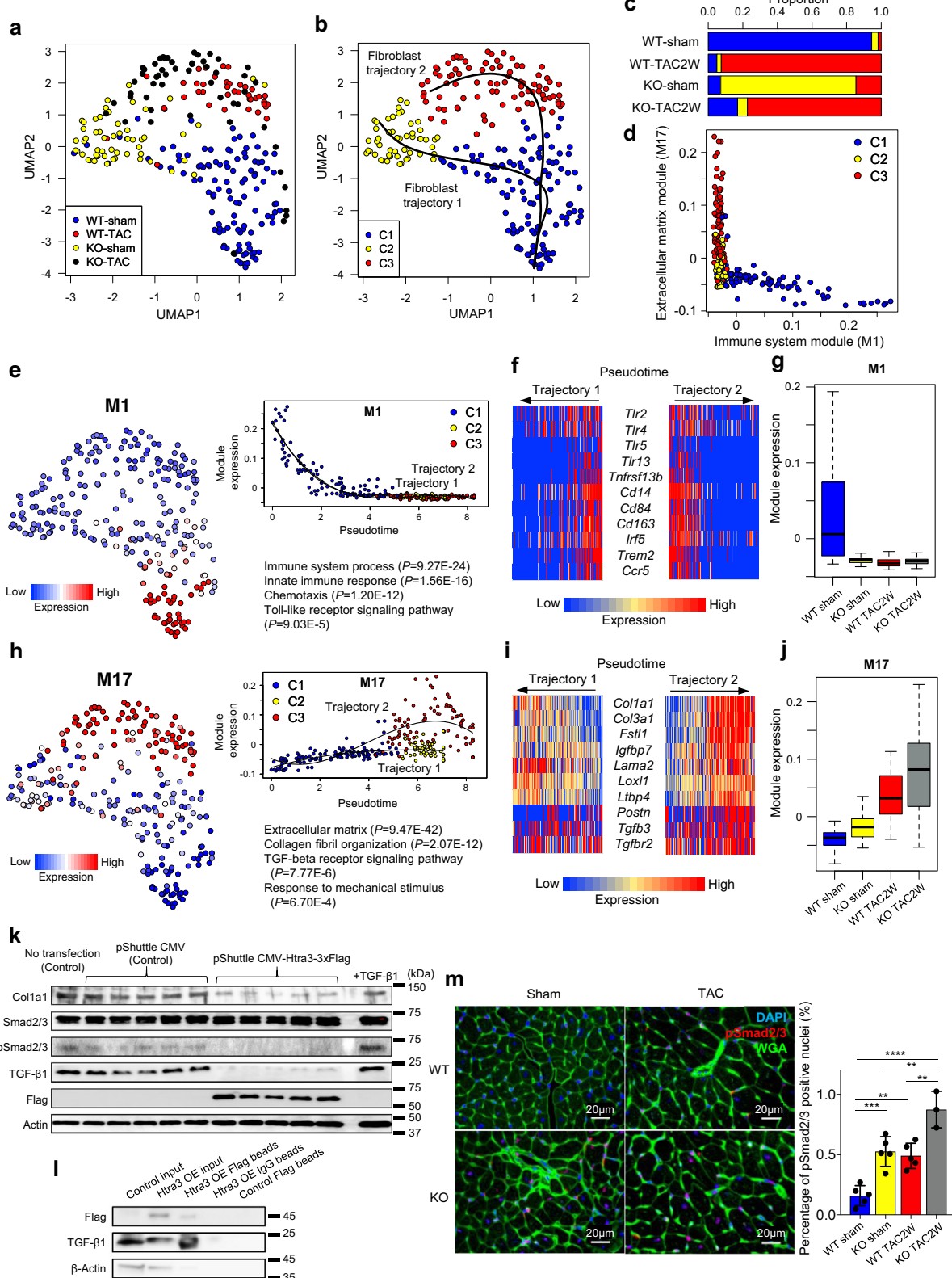

mitochondria (e.g., mt-Nd6 and mt-Atp6), in the non-infarct zone both in WT and *Htra3* KO mice (Fig. 6d, e). Regions assigned to cluster 2, which corresponded to the infarct zone, were enlarged in *Htra3* KO mice after MI (Fig. 6d). Gene ontology analysis showed that cluster 2 was characterized by genes involved in several biological phenomena, such as

extracellular matrix organization, response to TGF-β, inflammatory response, and integrin-mediated signaling pathway (Supplementary Fig. 6b). We next conducted scRNA-seq of isolated cells from the infarct zone after MI (Supplementary Fig. 6c) and deconvoluted spatially located spots using scRNA-seq profiles, indicating that fibroblasts were highly enriched in regions

**Fig. 3 Single-cell RNA-seq of cardiac fibroblasts reveals that Htra3 maintains their quiescent state by TGF-β inhibition. a** UMAP plot of single-cell transcriptomes of cardiac fibroblasts isolated from WT and *Htra3* KO mice at 2 weeks after TAC or sham surgery using an antibody against PDGFR-α ($n = 109$ cells for WT sham, $n = 40$ for WT TAC, $n = 55$ for KO sham, $n = 53$ for KO TAC). **b** Trajectory analysis on the UMAP plot in **a**. Clusters classified by graph-based clustering are shown by colors. Trajectories identified by the Slingshot algorithm are also shown. **c** Bar graph showing the distribution of the clusters in **b**. **d** Scatter plot showing the module eigengene (ME) expression of M1 and M17 for each cell. **e** M1 expression on the UMAP plot (left) and its dynamics along pseudotime (upper right). Enriched GO terms and enrichment *P*-values are also shown (lower right). *p*-values are determined by Fisher's Exact test. **f** Heatmap showing the expression levels of selected M1 genes during the trajectories. **g** Boxplot of the M1 expression. Data represent box plots and individual data points. Box plots show the median (center line), first and third quartiles (box edges), while the whiskers going from each quartile to the minimum or maximum. $n = 109$ for WT-sham, $n = 55$ for KO-sham, $n = 40$ for WT-TAC, $n = 53$ for KO-TAC. **h** M17 expression on the UMAP plot (left) and its dynamics along pseudotime (upper right). Enriched GO terms are also shown (lower right). *p*-values are determined by Fisher's Exact test. **i** Heatmap showing the expression levels of selected M17 genes during the trajectories. **j** Boxplot of the M17 expression. Data represent box plots and individual data points. Box plots show the median (center line), first and third quartiles (box edges), while the whiskers going from each quartile to the minimum or maximum. $n = 109$ for WT-sham, $n = 55$ for KO-sham, $n = 40$ for WT-TAC, $n = 53$ for KO-TAC. **k** Western blot analysis using cardiac fibroblasts. TGF-β1 treatment was used as a positive control for pSmad2/3. **l** Immunoprecipitation followed by western blot analysis showing the binding of Htra3 to mature TGF-β1. **m** Immunostaining of pSmad2/3 on heart sections. pSmad2/3(Red), Wheat germ agglutinin (WGA, Green) and 4′,6-diamidino-2-phenylindole (DAPI, Blue) are used to stain the plasma membrane and nucleus, respectively. Results of quantitative analysis are also shown ($n = 5, 5, 5, 3$ biologically independent mice, from left to right). Data are shown as mean and SD. **$P < 0.01$, ***$P < 0.005$, ****$P < 0.001$; Significance was determined by one-way ANOVA with Tukey's or Dunnett's post hoc test. Source data are provided as a Source Data file.

assigned to cluster 2 (Fig. 6f). After MI, the fibroblast population was divided into 4 clusters (Supplementary Fig. 6c). Fibroblast clusters 2 (FB2) and 4 (FB4), which were increased in the early phase after myocardial infarction, showed lower expression levels of *Htra3* (Supplementary Fig. 6d–f) and the total expression levels of *Htra3* were decreased at the early phase after MI (Supplementary Fig. 6g), which are consistent with the finding that expression of *Htra3* in cardiac fibroblasts was down-regulated after pressure overload to the heart (Fig. 2e) or mechanical stretch (Fig. 2g). We also conducted trajectory analysis of cardiomyocytes after MI using scRNA-seq profiles to reveal that MI induced failing cardiomyocytes, which are characterized by the expression of TGF-β signaling-related molecules and secretory factors (e.g., *Bmp1*, *Tgfb3*, and *Igfbp7*) (Fig. 6g, h, Supplementary Fig. 6h, i). Spatial transcriptomic analysis demonstrated that these genes, together with extracellular matrix genes, were specifically activated at the infarct zone in *Htra3* KO mice after MI (Fig. 6i, Supplementary Fig. 6j), suggesting that Htra3 downregulation promotes cardiac fibrosis and induces cardiomyocyte secretory phenotype in infarct regions.

**Transcriptomic signatures of failing cardiomyocytes are conserved in human heart failure.** To assess the conservation of transcriptomic signatures in human failing cardiomyocytes, we performed full-length scRNA-seq of cardiomyocytes isolated from patients with heart failure ($n = 22$) and control subjects ($n = 2$). UMAP plotting and graph-based clustering classified cardiomyocytes into 3 clusters, and pseudotime analyses revealed 2 trajectories (Fig. 7a, b, Supplementary Fig. 7a). Trajectories 1 and 2 corresponded to the transition from C1 to C2 and C1 to C3, respectively (Fig. 7b). Cardiomyocytes from control subjects mainly belonged to C1, whereas those from patients with heart failure belonged to C2 or C3 (Fig. 7c). Random forest and overlap analysis of identified co-expression gene modules identified modules M1, M2, and M44 as significantly involved in cell classification and conserved between human and mouse (Supplementary Fig. 7b–d).

Module M2, which corresponded to murine cardiomyocyte M1 (Supplementary Fig. 7d) and was enriched with genes involved in mitochondrial electron transport and ATP biosynthesis (Fig. 7d), was suppressed during both trajectories (Fig. 7d–f). By contrast, module M1, which corresponded to murine cardiomyocyte M2 (Supplementary Fig. 7d) and was enriched with genes involved in type I interferon signaling (e.g., *IFNAR1*, *IRF9*, and *JAK1*), TGF-β receptor signaling (e.g., *TGFB3* and *TGFBR2*), small GTPase

signaling (e.g., *RHOA* and *RAC1*), and DNA damage response (e.g., *CDKN1A*, *MDM4*, and *RBX1*), was activated specifically in trajectory 2 (Fig. 7g–i, Supplementary Fig. 7e), in which M1 and M2 showed a mutually exclusive relationship in their module activities (Fig. 7j). M1 also contained secretory factors specifically expressed in murine senescent failing cardiomyocytes (e.g., *CXCL12* and *IGFBP7*) (Fig. 7h, Supplementary Fig. 4g). By calculating the overlap between gene modules detected from single-cardiomyocyte RNA-seq in human and mice, we also found that M44, which corresponded to murine cardiomyocyte M9 (Supplementary Fig. 7d), was enriched with genes involved in extracellular matrix organization and activated specifically in trajectory 2 (Fig. 7k–m). This was confirmed by single-molecule RNA in situ hybridization (Supplementary Fig. 7f). These results suggest that human failing cardiomyocytes show the same conserved features as murine failing cardiomyocytes, including impaired mitochondrial biogenesis, activated TGF-β receptor signaling, DNA damage response, and fibroblast-like secretory phenotype.

**IGFBP7, secreted from failing cardiomyocytes, is identified to be a biomarker for progression of heart failure.** To clarify whether cytokines secreted from failing cardiomyocytes might be linked to the pathogenesis of heart failure, we performed plasma proteome analysis on human heart failure at different stages ($n = 768$ for control subjects, $n = 84$ for patients with non-advanced heart failure, $n = 30$ for patients with advanced heart failure both before and after heart transplantation)[34]. Levels of cytokines such as TGF-β3, LTBP4, and IGFBP7, which were expressed from senescent failing cardiomyocytes, were high in patients with heart failure (depending on the severity) and low after transplantation, similar to NT-proBNP, a well-established biomarker of heart failure[35] (Fig. 7n, Supplementary Fig. 7g). Random forest analysis identified IGFBP7 as the protein most strongly involved in the classification between non-advanced and advanced heart failure (Fig. 7o). Receiver-operating characteristic analysis demonstrated that NT-proBNP had better ability to diagnose heart failure, whereas IGFBP7 had a higher power to determine the severity of heart failure (Fig. 7p, q). These results suggest that IGFBP7, secreted from failing cardiomyocytes, is a useful biomarker for advanced heart failure in humans.

**Discussion**

In this study, we demonstrate that cardiac fibroblasts are critically involved in the development of heart failure by inducing not only

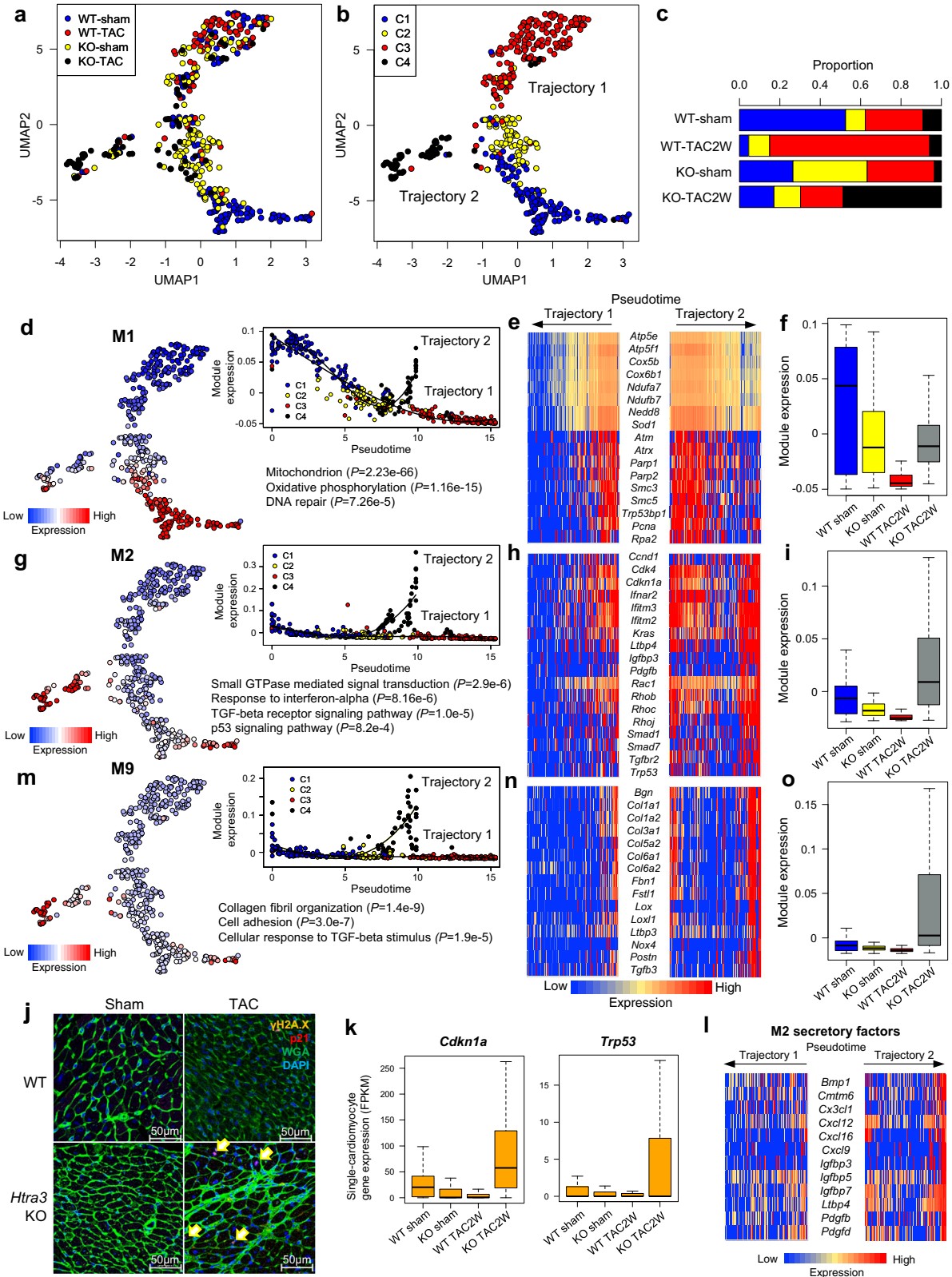

cardiac fibrosis but also cardiomyocyte secretory phenotype through Htra3-TGF-β-IGFBP7 axis. Mechanical stress on fibroblasts reduces expression of Htra3, leading to upregulation of TGF-β in the heart. Sustained activation of TGF-β signaling represses expression of many DNA repair genes in cardiomyocytes, consistent with the previous findings observed in cancer and bone marrow failure that TGF-β signaling-induced accumulation of DNA damage is based on transcriptional repression of DNA repair genes[36,37]. Sustained activation of TGF-β signaling also activates expression of Nox4, a major source of oxidative stress in the failing heart[33], and promotes ROS production in cardiomyocytes. These events collectively induce accumulation of

**Fig. 4 Single-cell RNA-seq of cardiomyocytes shows that Htra3 prevents the induction of senescent failing cardiomyocytes. a** UMAP plot of single-cell transcriptomes of cardiomyocytes isolated from WT and *Htra3* KO mice 2 weeks after TAC or sham surgery ($n = 175$ cells for WT sham, $n = 87$ for WT TAC, $n = 117$ for KO sham, $n = 75$ for KO TAC). **b** Trajectory analysis on the UMAP plot in **a**. Clusters classified by graph-based clustering are shown by colors. Trajectories identified by the Slingshot algorithm are also shown. **c** Bar graph showing the distribution of the clusters in **b**. **d** M1 expression on the UMAP plot (left) and its dynamics along pseudotime (upper right). Enriched GO terms and enrichment *P*-values are also shown (lower right). *p*-values are determined by Fisher's Exact test. **e** Heatmap showing the expression levels of selected M1 genes during the trajectories. **f** Boxplot of the M1 expression. Data represent box plots and individual data points. Box plots show the median (center line), first and third quartiles (box edges), while the whiskers going from each quartile to the minimum or maximum. $n = 175$ for WT-sham, $n = 117$ for KO-sham, $n = 87$ for WT-TAC, $n = 75$ for KO-TAC. **g** M2 expression on the UMAP plot (left) and its dynamics along pseudotime (upper right). Enriched GO terms are also shown (lower right). *p*-values are determined by Fisher's Exact test. **h** Heatmap showing the expression levels of selected M2 genes during the trajectories. **i** Boxplot of the M2 expression. Data represent box plots and individual data points. Box plots show the median (center line), first and third quartiles (box edges), while the whiskers going from each quartile to the minimum or maximum. $n = 175$ for WT-sham, $n = 117$ for KO-sham, $n = 87$ for WT-TAC, $n = 75$ for KO-TAC. **j** Immunostaining of γH2A.X (Yellow) and p21 (Red) on heart sections from WT and *Htra3* KO mice after TAC or sham surgery (4 weeks). WGA (Green) and DAPI (Blue) are used to stain the plasma membrane and nucleus, respectively. Arrows indicate the gH2A.X/p21-positive nuclei. **k** Boxplot showing expression of DNA damage-related genes (Cdkn1a and Trp53) in single-cardiomyocytes. Data represent box plots and individual data points. Box plots show the median (center line), first and third quartiles (box edges), while the whiskers going from each quartile to the minimum or maximum. $n = 175$ for WT-sham, $n = 117$ for KO-sham, $n = 87$ for WT-TAC, $n = 75$ for KO-TAC. **l** Heatmap showing the expression levels of selected M2 secretory factor genes during the trajectories. **m** M9 expression on the UMAP plot (left) and its dynamics along pseudotime (upper right). Enriched GO terms are also shown (lower right). *p*-values are determined by Fisher's Exact test. **n** Heatmap showing the expression levels of selected M9 genes during the trajectories. **o** Boxplot of the M9 expression. Data represent box plots and individual data points. Box plots show the median (center line), first and third quartiles (box edges), while the whiskers going from each quartile to the minimum or maximum. $n = 175$ for WT-sham, $n = 117$ for KO-sham, $n = 87$ for WT-TAC, $n = 75$ for KO-TAC.

DNA damage and activation of p53 signaling, resulting in the induction of failing cardiomyocytes with expression of TGF-β signaling-related molecules and secretory factors, similar to senescence in proliferative cells[2,38,39].

TGF-β signaling and DNA damage are involved in many aging-related disorders, including neurodegenerative diseases[40] and cancers[41]. Since DNA damage and subsequent cellular senescence activate expression of TGF-β signaling-related molecules, which plays a major role in paracrine senescence[42–44], the vicious cycle between TGF-β signaling and DNA damage may be the pathological basis of heart failure as well as these aging-related disorders. Since Htra3 expression was downregulated in cardiac fibroblasts in the murine and human failing heart and *Htra3* overexpression in the murine heart inhibited TGF-β signaling and ameliorated cardiac dysfunction as well as fibrosis (Fig. 5h–j), Htra3-mediated prevention of cardiac fibrosis and cardiomyocyte secretory phenotype would be a novel therapeutic strategy for heart failure, in addition to senolytic therapies[18,45]. Htra3 expression has been shown to be down-regulated with increasing grades in endometrial and ovarian cancer[46,47], suggesting that Htra3 might also be a target of cancer treatment.

Our integrative analyses of single-cardiomyocyte transcriptome and plasma proteome in human identified a clinically significant link between failing cardiomyocytes and cytokines including IGFBP7. It has been reported that expression of IGFBP7 is upregulated by TGF-β via smad2[48] and that TGF-β-induced IGFBP7 expression in the stroma defines poor-prognosis subtypes in colorectal cancer[49]. IGFBP7 has been reported to be secreted from senescent cells and to have the ability to induce senescence[50], suggesting that IGFBP7 itself might be involved in the development of heart failure. Context dependent SASPs have been reported to affect tumor microenvironment and immune response[51,52] and to be possible therapeutic targets[53,54]. Similarly, failing cardiomyocyte-specific secretory phenotype and its secreted cytokine IGFBP7 might be potential therapeutic targets for heart failure.

Despite the significance of the findings, our study has several limitations. First, we did not use fibroblast-specific *Htra3* KO mice. However, since Htra3 is predominantly expressed in cardiac fibroblasts (Fig. 1i, Supplementary Figs. 1j, k, 6, 6f), our global *Htra3* KO mice are considered to be well enough for the evaluation of the functional significance of Htra3 in the heart. Second, in

single-cardiomyocyte RNA-seq of human samples, the difference in the time from death (for control subjects) or tissue sampling (for patients with heart failure) to cell isolation may generate some bias in transcriptomes. However, we set the same cutoff for all scRNA-seq data for subsequent analysis in quality control and detected gene modules corresponded to gene modules detected from murine single-cardiomyocyte RNA-seq. In addition, we identified some secretory factors which are secreted from failing cardiomyocytes and are associated with the severity of heart failure by integrating with plasma proteome analysis. Our dataset would advance further research regarding the pathogenesis of heart failure.

## Methods

**Animal model**. C57BL/6N mice were purchased from CLEA Japan, Inc. *Htra3* KO mice were generated by the conventional homologous recombination method using RF8 ES cells as well as the previous generation of *Htra1* KO mice[55]. DT-Ap/Neo cassette was kindly provided from Riken Bioresource Research Center, Japan. The targeting vector was designed to delete 424 base pairs from the 5′-end of exon 1 of the *Htra3* gene. (Supplementary Fig. 2a) To validate successful gene targeting in ES cells, 10 µg genomic DNA extracted from ES cells was digested with *Bam*HI and hybridized with 800-bp DIG-labelled probe, which was derived from the region near intron 2 of *Htra1* gene (Supplementary Fig. 2a). The absence of the *Htra3* transcript in homozygous *Htra3* KO mice was confirmed by northern blot analysis. Total RNA (20 µg) prepared from the heart was hybridized with 1.4 kbp DIG-labelled full-length cDNA of *Htra3* (Supplementary Fig. 2a). *Htra3* KO mice were maintained as the C57BL/6 N background after more than 10 generations of backcrossing. Only male mice were used in all experiments. Female and male mice were housed in separate cages at a maximum of 6 mice per cage in a specific-pathogen–free, temperature-controlled vivarium under a 12-h light/dark cycle with ad libitum access to food and water. Ambient room temperature was regulated at $73 \pm 5$ °F and humidity was controlled at $50 \pm 10\%$. All experiments were approved by the University of Tokyo Ethics Committee for Animal Experiments and strictly adhered to the guidelines of the University of Tokyo for animal experiments. *Htra3* KO mice are available with the material transfer agreement.

**Systolic blood pressure measurement**. Systolic blood pressure was measured in conscious mice (WT $n = 12$ vs Htra3 KO $n = 12$) by the tail-cuff system using MK-2000ST (Muromachi Kikai, Tokyo, Japan) according to the manufacturer's protocol.

**Operation of TAC model and echocardiography**. TAC surgery was performed in male mice randomly selected from among those weighing 20–23 g at the age of 9–11 weeks as previously described[56]. After the mice were anaesthetized with 2% isoflurane via inhalation, the aorta was approached via minimal sternal incision and a 7-0 ligature was placed around the vessel using a 26-gauge needle to ensure consistent occlusion. Sham-operated animals, which underwent a similar surgical

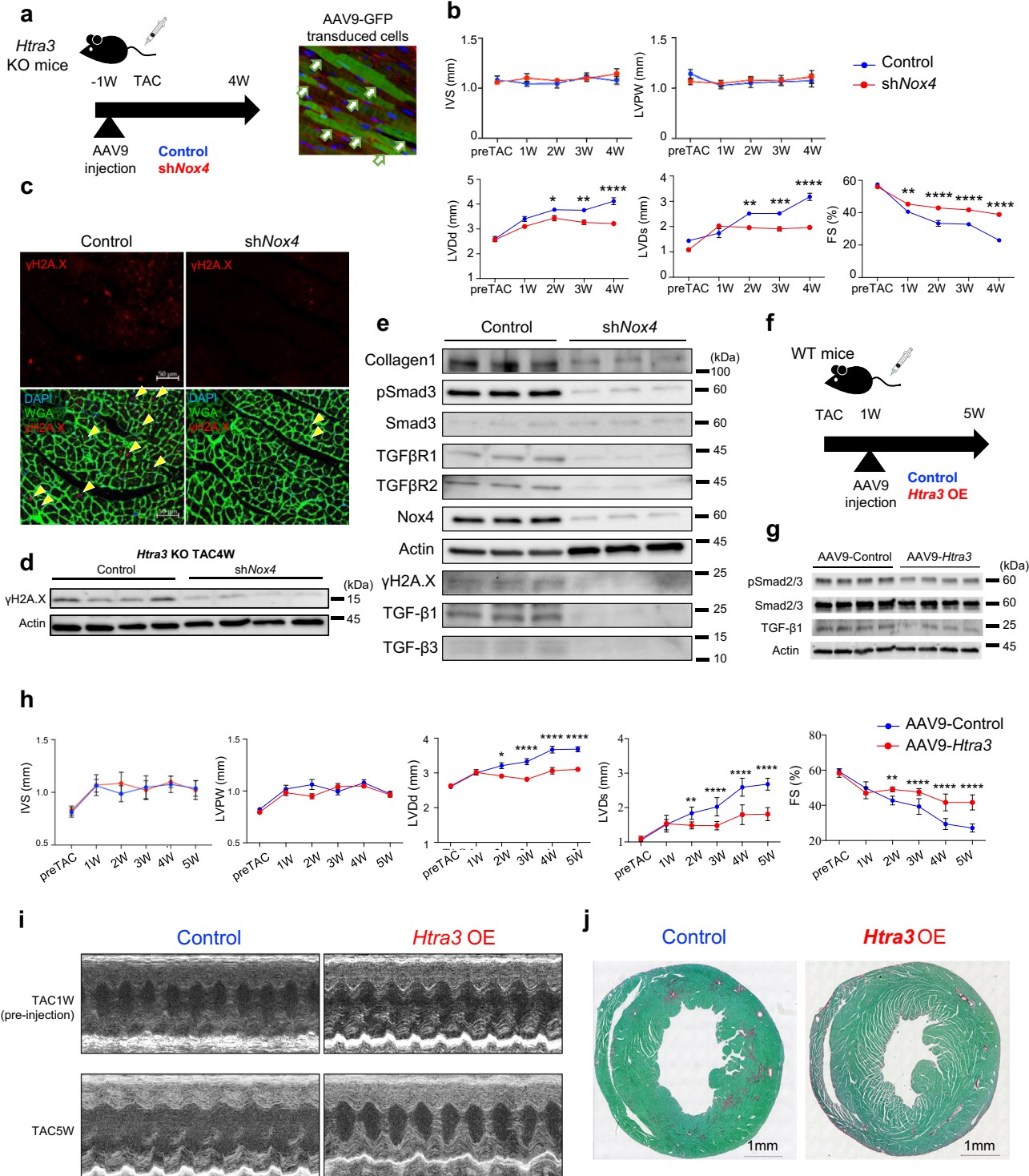

procedure without aortic constriction, were used as controls. The surgeon was not informed about the genotypes of the mice. Mice that died within 1 week after the operation were excluded from the analysis. Transthoracic echocardiography was performed on conscious mice, using a Vevo 2100 imaging system (Visualsonics, Inc.). To minimize variation in the data, cardiac function was assessed only when the heart rate was 600–700 beats per minute. M-mode echocardiographic images were obtained from a longitudinal view to measure the size and function of the left ventricle.

develop MI or died within one week after the operation were excluded from the analysis.

**Single-cell RNA-seq analysis of mouse samples**. For the isolation and collection of non-cardiomyocytes, hearts were minced and enzymatically dissociated using 2 mg/mL type 2 collagenase (Worthington), 1 mg/mL dispase (Roche), and 20 U/mL DNAse I (Roche), with 5 cycles of digestion for a total 40 min at 37 °C. After using a 40 μm cell strainer (Greiner) to remove the cardiomyocytes, cells were stained with Zombie Green Fixable Viability Kit (BioLegend) and live cells were collected by fluorescence-activated cell sorting (FACS) using a FACSJazz cell sorter (BD Biosciences). To isolate cardiac fibroblasts, cells were stained with APC conjugated anti-mouse CD140a (PDGFR-α) antibody (#135907, BioLegend, 1:200),

**Myocardial infarction model**. Myocardial infarction (MI) was induced as previously described[57]. Mice were anaesthetized by inhalation of 2% isoflurane. MI was induced by ligation of the left anterior descending artery. Mice that failed to

**Fig. 5 TGF-β-induced Nox4 activation induces senescent failing cardiomyocytes and heart failure. a** Experimental design for testing the effect of sh*Nox4* AAV9 vectors and validation of AAV9 transduction with the AAV9-eGFP vector. WGA (Red) and DAPI (Blue) are used to stain the plasma membrane and nucleus, respectively. Arrows indicate the GFP transduced cells. **b** Echocardiographic assessment of the heart from TAC-induced *Htra3* KO mice after injection of control or sh*Nox4* AAV9 vectors ($n = 4$ each). Data are shown as mean and SD. *$P < 0.05$, **$P < 0.01$, ****$P < 0.001$; significance was determined by two-way ANOVA with Bonferroni's multiple comparison test. Source data are provided as a Source Data file. **c** Immunostaining of γH2A.X on the heart sections from TAC-induced *Htra3* KO mice after injection of control or sh*Nox4* AAV9 vectors. WGA (Green) and DAPI (Blue) are used to stain the plasma membrane and nucleus, respectively. Arrows indicate the γH2A.X-positive nuclei. **d** Western blot analysis of γH2A.X using isolated cardiomyocytes from TAC-induced *Htra3* KO mice after injection of control or sh*Nox4* AAV9 vectors. **e** Western blot analysis using isolated cardiomyocytes from TAC-induced *Htra3* KO mice after injection of control or shNox4 AAV9 vectors. **f** Experimental design for testing the effect of AAV9-*Htra3* vectors. **g** Western blot analysis using isolated cardiomyocytes from TAC-induced mice after injection of AAV9-Control or AAV9-*Htra3* vectors. **h** Echocardiographic assessment of the heart from TAC-induced mice after injection of AAV9-Control or AAV9-*Htra3* vectors ($n = 7$ each). Data are shown as mean and SD. *$P < 0.05$, **$P < 0.01$, ****$P < 0.001$; significance was determined by two-way ANOVA with Bonferroni's multiple comparison test. Source data are provided as a Source Data file. **i** Representative echocardiographic images of WT mice injected with either AAV9-Control or AAV9-*Htra3*. **j** Histochemical detection of collagen fibers by Sirius Red/Fast Green dye staining of the heart from TAC-induced mice after injection of AAV9-Control or AAV9-*Htra3* vectors.

FITC conjugated anti-mouse CD31 Antibody (#102506, BioLegend, 1:200), and Zombie Green, after which PDGFR-α positive live cells were collected by FACS.

Cardiomyocytes were isolated using Langendorff perfusion from the left ventricular free wall. Enzymatic dissociation using Langendorff perfusion was performed with 35 mL enzyme solution (type 2 collagenase 1 mg/mL [Worthington], protease type XIV 0.05 mg/mL [Sigma-Aldrich], NaCl 130 mM, KCl 5.4 mM, MgCl₂ 0.5 mM, NaH₂PO₄ 0.33 mM, D-glucose 22 mM, HEPES 25 mM; pH 7.4) pre-heated to 37 °C, at a rate of 3 mL/min. To neutralize enzymatic activity, foetal bovine serum (FBS) was added to the solution to a final concentration of 0.2% (v/v). Cell suspensions were filtered through a 100-μm nylon mesh cell strainer and centrifuged at 20 g for 3 min. The supernatant includes non-cardiomyocytes and was able to be collected by 300 g × 5 min centrifuging. To prevent hypercontraction, the cardiomyocytes were resuspended in medium (NaCl 130 mM, KCl 5.4 mM, MgCl₂ 0.5 mM, NaH₂PO₄ 0.33 mM, D-glucose 22 mM, HEPES 25 mM, FBS 0.2%; pH 7.4) containing a low concentration of calcium (0.1 mM). Rod-shaped live cardiomyocytes (viability of cardiomyocytes was ≥80% for all time points) were collected immediately after isolation from 2 mice with a 0.2–2 μL pipette (sample volume, 0.5 μL) and incubated in lysis buffer.

Single-cell cDNA libraries were generated using the Smart-seq2 protocol[12,21] or the Chromium 3′ v2 chemistry kit (10x Genomics) according to the manufacturer's instruction. For the cDNA libraries constructed by Smart-seq2, the efficiency of reverse transcription was assessed by examining the cycle threshold (Ct) values of control genes (*Tnnt2* for cardiomyocytes and *Actb* for non-cardiomyocytes) from quantitative real-time polymerase chain reaction (qRT-PCR) using a CFX96 Real-Time PCR Detection System (Bio-Rad), and the distribution of cDNA fragment lengths was assessed using LabChip GX (Perkin Elmer) and/or TapeStation 2200 (Agilent Technologies). The following primer sets were used for qRT-PCR: *Tnnt2* mRNA forward: TCCTGGCAGA GAGGAGGAAG; *Tnnt2* mRNA reverse: TGCAGGTCGA ACTTCTCAGC; *Actb* mRNA forward: CAACTGGGACGACATGGAGA; and *Actb* mRNA reverse: GCATACAGGGACAGCACAGC. A Ct value of 25 was set as the threshold. The sequencing libraries were subjected to paired-end 51-bp RNA sequencing on a HiSeq 2500 (Illumina) in rapid mode. Insert size was 345 ± 40 bp (average ± standard deviation). RefSeq transcripts (coding and non-coding) were downloaded from the UCSC Genome Browser (http://genome.ucsc.edu). Reads were mapped to the mouse genome (mm9) with the parameters "-g 1 -p 8 mm9— no-coverage-search" using TopHat and Cufflinks[58,59]. FPKM was calculated with reads mapped to the nuclear genome[60]. In single-cell RNA-seq through full-length cDNA library synthesis by Smart-seq2, we calculated detected genes for each cell and generated histogram to set the cutoffs for genes to be analyzed.

For weighted co-expression network analysis, all genes expressed at an FPKM value of ≥10 in at least one of the samples were used to construct a signed network using the WGCNA R package[23], which was also used for cell type annotation. The soft power threshold was analyzed with the "pickSoftThreshold" function and applied to construct a signed network and calculate the module eigengene expression using the "blockwiseModules" function. Modules with <30 genes were merged with their closest larger neighboring module. To visualize the weighted co-expression networks, Cytoscape (version 3.7.2)[61] with "edge-weighted force-directed" was used. The correlation coefficient between each gene expression with fibroblast module expression was calculated in Fig. 3h and Supplementary Table 2. For dimension reduction, UMAP[62] was used. Graph-based clustering was performed using the "buildSNNGraph" function in scran[63]. To assess the accuracy of the classification, the "randomForest" package in R was used. The cell-type of each cluster was determined by characteristic marker genes such as *Myl2*, *Myl4*, *Myh6*, and *Myh7* for cardiomyocyte, *Kdr*, *Fabp4*, and *Vwf* for endothelial cell, *Col1a1*, *Dcn*, and *Lum* for fibroblast, whose expression profiles were shown in Supplementary Figs. 1a, 2f. and For LR communication analysis, a published database[22] was used to extract LR interaction pairs. The matrix data regarding the LR communication in the heart is provided in Supplementary Table 1. For

pseudotime analysis, Slingshot[64] was used to detect trajectories. Pearson's correlation coefficient was calculated using the "cor" function in R. Single-cardiomyocyte RNA-seq data of p53 KO mice were download from the Gene Expression Omnibus with the accession code GSE95143. Hierarchical clustering was performed using Cluster 3.0[65] and JAVA Treeview[66].

Regarding the droplet-based scRNA-seq data using Chromium Controller (10x Genomics) (Fig. 1a), the dataset was aligned and quantified using the CellRanger software package with default parameters. Cells with >300 expressed genes were retained, resulting in 1783 cells from 2 mice (sham operation) for the subsequent analysis using Seurat v3[67]. SCTransform normalization was implemented separately for each dataset, after which the top 500 feature genes were selected and used for data integration. The integrated data with the 'FindIntegrationAnchors' function was used for dimensionality reduction and cluster detection. Marker genes specific for each cluster were used for cell type annotation.

**Gene ontology analysis.** The Database for Annotation, Visualization, and Integrated Discovery (DAVID)[68] was used for GO analysis and Kyoto Encyclopedia of Genes and Genomes (KEGG) pathway enrichment analysis. Enrichment *P*-values of all extracted GO terms for each module were calculated in DAVID.

**Spatial transcriptome analysis of mouse MI model.** Frozen samples were embedded in OCT (TissueTek Sakura) and cryosectioned at −14 °C (Leica CM1860). Sections were placed on chilled Visium Spatial Gene Expression Slides (2000233, 10X Genomics), and adhered by warming the back of the slide. Tissue sections were then fixed in chilled methanol and stained according to the Visium Spatial Gene Expression User Guide (CG000239 Rev D, 10X Genomics). For gene expression samples, tissue was permeabilized for 15 min, which was selected as the optimal time based on tissue optimization time course experiments (CG000238 Rev D, 10X Genomics). Brightfield histology images were taken using a 10X objective on a BZ-X700 microscope (Keyence Corporation, Itasca, Illinois). Raw images were stitched together using BZ-X analyzer software (Keyence Corporation) and exported as TIFF files.

Libraries were prepared according to the Visium Spatial Gene Expression User Guide (CG000239 Rev D, 10X Genomics) and sequenced on a NovaSeq 6000 System (Illumina) using a NovaSeq S4 Reagent Kit (200 cycles, 20027466, Illumina), at a sequencing depth of approximately 250–400 M read-pairs per sample. Sequencing was performed using the following read protocol: read 1, 28 cycles; i7 index read, 10 cycles; i5 index read, 10 cycles; read 2, 91 cycles. Raw FASTQ files and histology images were processed by sample with the Space Ranger software (ver 1.1.0, 10X Genomics), against the Cell Ranger mm10 reference genome "refdata-gex-mm10-2020-A", available at: https://cf.10xgenomics.com/ supp/spatial-exp/refdata-gex-mm10-2020-A.tar.gz. Seurat v4 was used for downstream analysis. SCTransform normalization was implemented separately for each dataset and merge them. The merged data were used for dimensionality reduction and cluster detection. Differentially expressed genes were detected by using FindMarkers in Seurat (log2fc.threshold > 0.25, p_val_adj < 0.05). To predict the proportion of cell types in each spot, predict.score was calculated by using the 'FindTransferAnchors' and 'TransferData' functions in Seurat. The scRNA-seq of cells isolated from mice after sham or MI operation were used as reference data.

**Single-cell RNA-seq analysis of human samples.** The procedure for single-cell RNA-seq of cardiomyocytes isolated from patients with heart failure and from control subjects was approved by the ethics committee of the University of Tokyo (Approval No. G-10032). All procedures were conducted according to the Declaration of Helsinki, and written informed consent was obtained from all participants. Heart tissue was obtained within 1 hour after death from 2 patients of non-cardiac causes with normal cardiac function during autopsy or from 22 patients with severe heart failure during left ventricular assist device surgery or

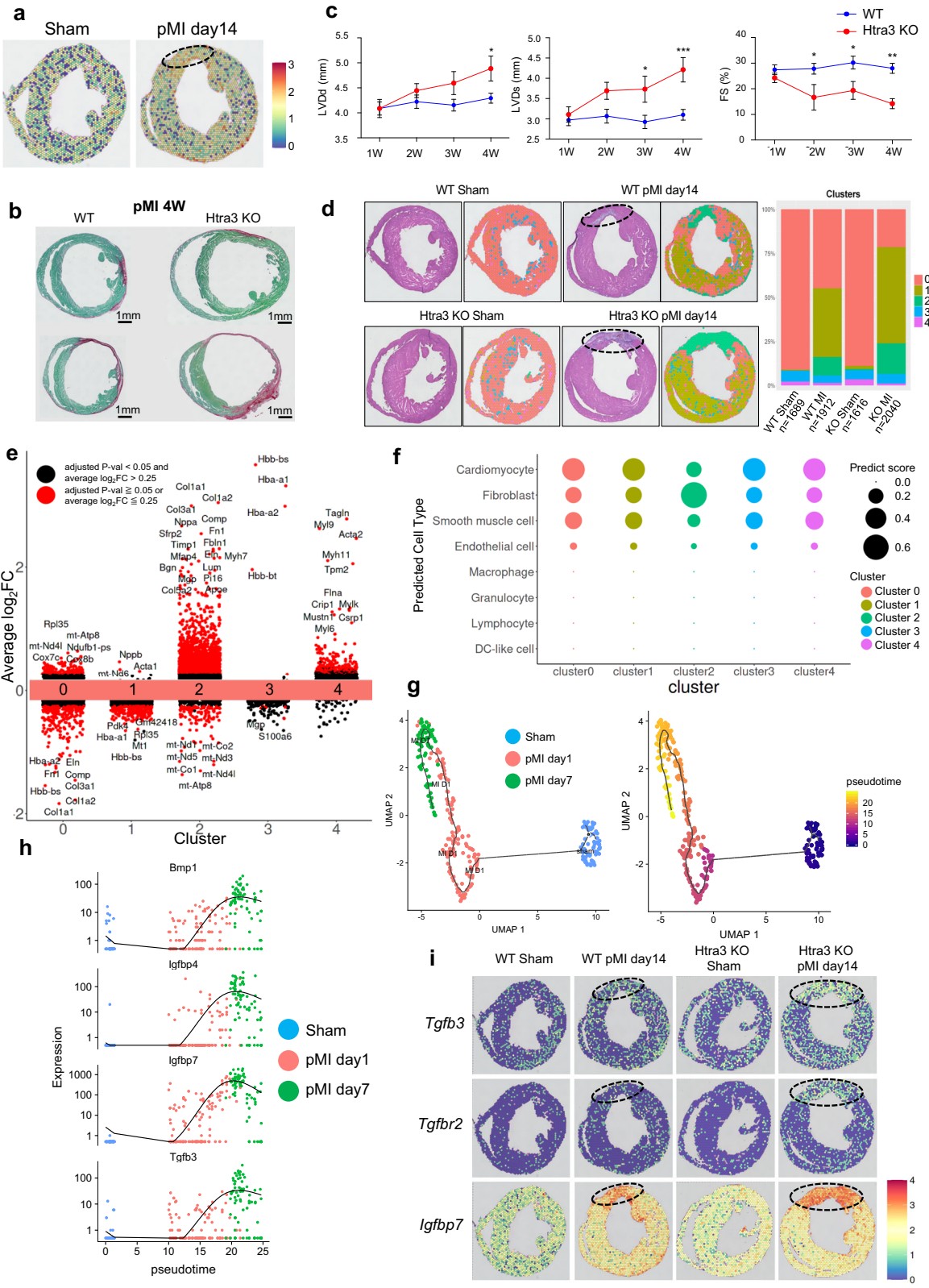

heart transplantation. The population including control subjects and heart failure patients contained 17 males and 7 females, and the mean age at the isolation of cardiomyocytes for scRNA-seq was 46.4 ± 12.2 years.

Immediately after the collection of the heart tissue, tissue was minced and incubated in lysis buffer containing 2 mg/mL type 2 collagenase (Worthington), 1 mg/mL dispase (Roche), and 20 U/mL DNAse I (Roche). After 4 cycles of lytic digestion by mild shaking for a total 20 min at 37 °C, rod-shaped live

cardiomyocytes were isolated. Single-cell cDNA libraries were generated using the Smart-seq2 protocol[21]. The efficiency of reverse transcription was assessed by examining the Ct values of a control gene (*TNNT2*) from qRT-PCR using a CFX96 Real-Time PCR Detection System (Bio-Rad). The distribution of the lengths of cDNA fragments was assessed using a LabChip GX (Perkin Elmer) and/or TapeStation 2200 (Agilent Technologies). The following primer set was used for qRT-PCR: *TNNT2* mRNA forward, AAGTGGGAAG AGGCAGACTGA; *TNNT2*

**Fig. 6 Spatial transcriptomic analysis reveals Htra3 repression-mediated spatial induction of senescent failing cardiomyocytes. a** Spatial expression profiles of *Htra3* across heart tissue sections of mice after sham or MI operation. The dashed circle represents the infarct region. **b** Histochemical detection of collagen fibers by Sirius Red/Fast Green dye staining in WT and *Htra3* KO mice at 4 weeks after myocardial infarction (MI) operation. **c** Echocardiographic assessment of the heart of wild-type (WT) and *Htra3* KO mice after MI operation. $n = 12$ (WT), $n = 6$ (KO). Data are shown as mean and SD. *$P < 0.05$, **$P < 0.01$, ***$P < 0.005$; significance was determined by two-way analysis of variance (ANOVA) with Bonferroni's multiple comparison test. **d** Hematoxylin and eosin stained tissue sections of the infarcted hearts and their corresponding spatial distribution of clusters characterized by specific expression profiles (left). The dashed circles represent the infarct regions. The colors of spatial points were corresponded to the bar graphs showing the distribution of clusters in each section (right). The number of analyzed spots are also shown. **e** Differential gene expression analysis showing up- and down-regulated genes across all five clusters. An adjusted *p*-value < 0.05 and log$_2$FC > 0.25 is indicated in red, while others are indicated in black. Source data are provided as a Source Data file. **f** Dot plot representing the distribution of cell-types predicted based on the scRNA-seq of cells isolated from mice after sham or MI operation. Dot size represents the average of predict score of each spot in each cluster. **g** Single-cell trajectory analysis of cardiomyocytes isolated from the infarct zone. **h** The expression dynamics of genes characteristic for senescent failing cardiomyocytes along pseudotime. **i** The spatial expression patterns of genes characteristic for senescent failing cardiomyocytes. The dashed circles represent the infarct regions.

mRNA reverse, GTCAATGGCC AGCACCTTC. A Ct value of 25 was set as the threshold. The remaining libraries were sequenced using a HiSeq 2500 System (Illumina). Reads were mapped to the human genome (hg19) with the parameters "-S -m 1 -l 36 -n 2 hg19" using Bowtie[69], as previously described[12]. RPKM values were calculated with reads mapped to the nuclear genome using DEGseq[70]. Single-cell transcriptomes consisting of over 2,000 detected genes were used for subsequent analysis.

Seurat v3[67] was used for single-cell RNA-seq analysis of the human heart, using a publicly available dataset (GSE121893)[4]. Cells with >500 expressed genes were retained, resulting in 2,459 cells from 8 patients (6 patients with heart failure, 2 healthy control subjects) for subsequent analysis. SCTransform normalization was implemented separately for each dataset, after which the top 2,000 feature genes were selected and used for data integration. The integrated data were used for dimensionality reduction and cluster detection. Marker genes specific for each cluster were used for cell type annotation.

**Anti-TGF-β treatment**. For experiments with anti-TGF-β treatment, *Htra3* KO mice were intraperitoneally injected daily with 1.5 mg/kg body weight of isotype IgG1 control antibody (#MAB002, R&D Systems) or anti-TGFβ1 antibody (#MAB240, R&D Systems) as previously described[71]. Anti-TGF-β treatment was started 24 h before TAC operation.

**Cell culture and transfection of plasmid**. We isolated cardiac fibroblasts from the heart of 1-day-old C57BL/6 N (Takasugi Experimental Animal Supply Co.). Isolated cardiac fibroblasts were cultured in Dulbecco's modified Eagle's medium supplemented with 10% FBS at 37 °C in 5% CO$_2$. Plasmid cDNA encoding *Htra3*, which was tagged with FLAG at the C-terminus (pCMV-*Htra3*-flag), was prepared by VectorBuilder Inc. and transfected into these fibroblasts, using Lipofectamine 3000 (Thermo Fisher Scientific) according to the manufacturer's instructions. The transfected cells were cultured in medium containing 10% FBS serum for 48 h prior to analysis. Each experiment was repeated at least three times.

**Western blotting**. Heart tissue, purified cardiomyocytes, purified non-cardio-myocytes, or cultured cardiac fibroblasts were used as samples for western blotting in the present study. Samples were homogenized and lysed with RIPA buffer containing 10 mM Tris-HCl, 150 mM NaCl, 5 mM EDTA, 1% Triton X-100, 1% sodium deoxycholate, and 0.1% sodium dodecyl sulfate with protease and phosphatase inhibitor cocktails for 30 min on ice. Lysates were centrifuged at 15,000 × g for 30 min and the supernatants were used as whole-cell extracts. Total protein concentrations in the supernatants were measured using a bicinchoninic acid assay (Pierce BCA Protein Assay Kit; Thermo Scientific). For western blot analysis, extracted protein samples were separated on 5–20% Mini-PROTEAN TGX precast gradient gels (Bio-Rad) and transferred onto nitrocellulose membranes (BioRad). Membranes were blocked with 5% FBS in Tris-buffered saline plus 0.05% Tween and incubated overnight at 4 °C with primary antibodies, followed by horseradish peroxidase-conjugated secondary antibodies (Cell Signaling Technology) and ECL Plus (GE Healthcare). Immunoreactive signals were detected using a LAS 4000 analyzer (GE Healthcare).

The following antibodies were used for western blotting: rabbit monoclonal anti-pSmad2/3(Ser456/467) antibody (#8828, Cell Signaling Technology,1:1000), rabbit monoclonal anti-Smad3 (phospho S423 + S425) antibody (ab52903, Abcam, 1:1000), rabbit monoclonal anti-Smad2/3 antibody (#8685, Cell Signaling Technology, 1:1000), rabbit monoclonal anti-Smad3 antibody (ab40854, Abcam, 1:1000), rabbit monoclonal anti-Collagen1 antibody (ab138492, Abcam, 1:2000), rabbit monoclonal anti-TGF beta1 antibody (ab179695, Abcam, 1:1000), rabbit polyclonal anti-TGF beta1 antibody (ab92486, Abcam, 1:1000), horseradish peroxidase (HRP)-linked rabbit polyclonal anti-DDDDK-tag antibody (PM020-7, MBL, 1:1000), mouse monoclonal ANTI-FLAG M2 antibody (Sigma-Aldrich, #F1804, 1:1000), HRP-linked horse anti-rabbit IgG antibody (#7074, Cell Signaling Technology, 1:3000), HRP-linked horse anti-mouse IgG antibody (#7076, Cell

Signaling Technology, 1:3000), anti-TGFβReceptor I antibody (ab235178, abcam, 1:1000), anti-TGFβReceptor II antibody (SAB4502960, Sigma-Aldrich, 1:500), anti-TGF beta3 antibody (ab53727, Abcam, 1:1000), anti-Phospho-Histone H2A.X (Ser139) Monoclonal antibody (#MA1-2022, Invitrogen, 1:1000), anti-NADPH oxidase 4 antibody (ab109225, Abcam, 1:2000), anti-HtrA3 polyclonal antibody (NB600-1151, Novus Biologicals, 1:1000), anti-Cardiac Troponin I polyclonal antibody (ab47003, Abcam, 1:2000), and anti-Actin Monoclonal Antibody (#MA5-11869, Invitrogen, 1:5000).

**Protein purification of Htra3-flag**. After transfection of pCMV-Htra3-flag into the primary culture of murine cardiac fibroblasts, Htra3-flag protein was immunoprecipitated at 4 °C using anti-DDDDK-tag magnetic beads (MBL M185-11) from the lysate of the cells. For the isotype control, mouse IgG2a-magnetic beads (MBL M076-11) were used. Immunoprecipitates (using the same beads) from fibroblasts without transfection were also used as negative controls. After immunoprecipitation, each sample was washed and eluted according to the manufacturer's instructions (MBL).

**RNA in situ hybridization**. Both mouse and human tissues were fixed with G-Fix (Genostaff), embedded in paraffin on CT-Pro20 (Genostaff), using G-Nox (Genostaff) as a less toxic organic solvent than xylene, and sectioned at 5 µm. RNA in situ hybridization was performed with an ISH Reagent Kit (Genostaff) according to the manufacturer's instructions. Tissue sections were de-paraffinized with G-Nox and rehydrated through an ethanol series and phosphate-buffered saline (PBS). The sections were fixed with 10% neutral buffered formalin (10% formalin in PBS) for 30 min at 37 °C, washed in distilled water, placed in 0.2N HCl for 10 min at 37 °C, washed in PBS, treated with 4 µg/mL proteinase K (Wako Pure Chemical Industries) in PBS for 10 min at 37 °C, washed in PBS, and placed in a Coplin jar containing 1× G-Wash (Genostaff), equal to 1× saline-sodium citrate. Hybridization was performed with probes (250 ng/mL) in G-Hybo-L (Genostaff) for 16 h at 60 °C. After hybridization, the sections were washed in 1× G-Wash for 10 min at 60 °C and in 50% formamide in 1× G-Wash for 10 min at 60 °C. Next, the sections were washed twice in 1× G-Wash for 10 min at 60 °C, twice in 0.1× G-Wash for 10 min at 60 °C, and twice in TBST (0.1% Tween 20 in Tris-buffered saline) at room temperature. After treatment with 1× G-Block (Genostaff) for 15 min at room temperature, the sections were incubated with anti-DIG AP conjugate (Roche Diagnostics) diluted 1:2000 with G-Block (Genostaff; dilated 1/50) in TBST for 1 h at room temperature. The sections were washed twice in TBST and incubated in 100 mM NaCl, 50 mM MgCl2, 0.1% Tween 20, and 100 mM Tris-HCl (pH 9.5). Coloring reactions were performed with NBT/BCIP solution (Sigma-Aldrich) overnight and then washed in PBS. The sections were counterstained with Kernechtrot stain solution (Muto Pure Chemicals) and mounted with G-Mount (Genostaff). We used paired mirror cardiac sections to perform immunostaining of Pdgfra protein and in situ hybridization of *Htra3* mRNA in Fig. 1j.

**Single-molecule RNA in situ hybridization**. For single-molecule RNA fluorescent in situ hybridization (Supplementary Fig. 7f), the RNAscope system[72] (Advanced Cell Diagnostics) was used with a probe against human *LUM* mRNA (NM_002345.3, #494761). Frozen sections (10 µm) were fixed in PBS containing 4% paraformaldehyde for 5 min at room temperature, dehydrated by serial immersion in 50%, 70%, and 100% ethanol for 5 min at room temperature, and treated for 30 min at room temperature. The probe was then hybridized for 2 h at 40 °C, followed by RNAscope amplification, and co-stained with Alexa Fluor 488-conjugated wheat germ agglutin (W11261; Thermo Fisher Scientific; 1:200) and DAPI to detect the cell membranes and nuclei. Images were obtained as Z stacks using In Cell Analyzer 6000 (GE Healthcare).

**Tissue histology**. For histological analysis, mice were anaesthetized by isoflurane inhalation and sacrificed by cervical dislocation. The chest was opened, and the heart was flushed with cold PBS via cardiac apical insertion of a 25-gauge needle.

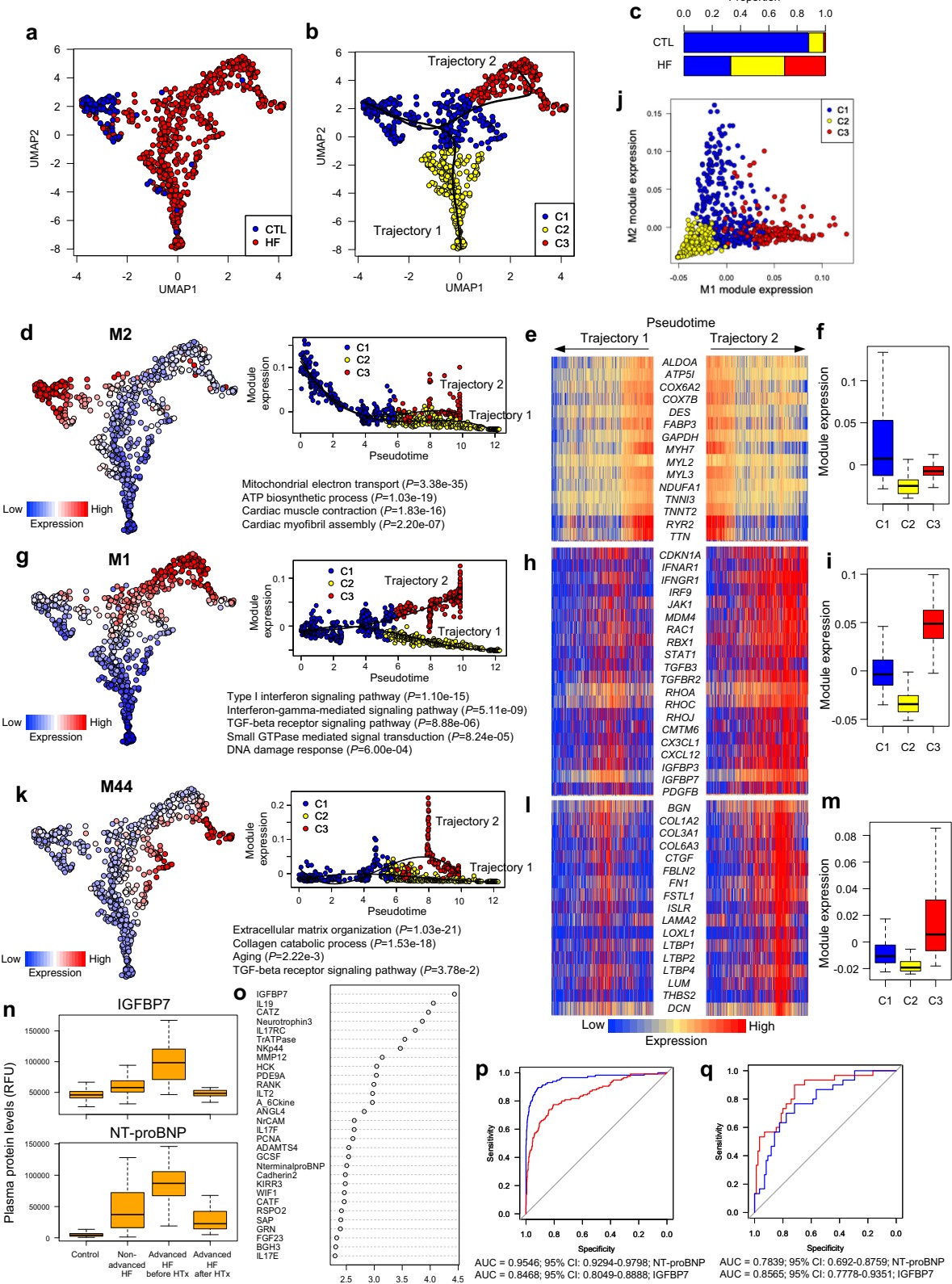

The right atrium was cut to allow drainage of blood from the heart, and the mice were briefly perfused with cold fixative (4% paraformaldehyde in PBS) through the apex of the heart. Tissues and organs were excised, flushed with fixative, incubated in fixative for 12 h at 4 °C with gentle rotation, and finally embedded in paraffin.

Paraffin-embedded heart tissues were sectioned into 4 μm slices using an SM2010 R Sliding Microtome (Leica Biosystems). For fluorescent immunostaining, the paraffin-embedded sections were treated with an antigen retrieval solution

(Dako) and incubated with primary antibodies overnight after blocking with 5% normal goat serum. After washing with PBS, samples were stained with appropriate secondary antibodies (anti-rabbit IgG-Alexa 594, 1:400; anti-mouse IgG-Alexa 647, 1:400) for 1 h. The membranes and nuclei of the cells were counterstained with Wheat Germ Agglutinin-Alexa 488 (Thermo Fisher Scientific; 1:200) and DAPI (4′,6-diamidino-2-phenylindole; Dojindo, 1:1000), respectively. Images were obtained using an LSM 880 META confocal microscope (Zeiss) or a BZ-X700

**Fig. 7 Single-cardiomyocyte RNA-seq and plasma proteome analysis of patients with heart failure. a** UMAP plot of single-cell transcriptomes of cardiomyocytes isolated from control subjects (CTL) or patients with heart failure (HF) ($n = 85$ cells for CTL, $n = 678$ for HF). **b** Trajectory analysis on the UMAP plot in **a**. Clusters classified by graph-based clustering are shown by colors. Trajectories identified by the Slingshot algorithm are also shown. **c** Bar graph showing the distribution of the clusters in **b**. **d** M2 expression on the UMAP plot (left) and its dynamics along pseudotime (upper right). Enriched GO terms are also shown (lower right). **e** Heatmap showing the expression levels of selected M2 genes during the trajectories. **f** Boxplot of the M2 expression. Data represent box plots and individual data points. Box plots show the median (center line), first and third quartiles (box edges), while the whiskers going from each quartile to the minimum or maximum. $n = 300$ for C1, $n = 267$ for C2, $n = 196$ for C3. **g** M1 expression on the UMAP plot (left) and its dynamics during trajectories (upper right). Enriched GO terms and enrichment $P$-values are also shown (lower right). **h** Heatmap showing the expression levels of selected M1 genes during the trajectories. **i** Boxplot of the M1 expression. Data represent box plots and individual data points. Box plots show the median (center line), first and third quartiles (box edges), while the whiskers going from each quartile to the minimum or maximum. $n = 300$ for C1, $n = 267$ for C2, $n = 196$ for C3. **j** Scatter plot showing the ME expression of M1 and M2 for each cell. **k** M44 expression on the UMAP plot (left) and its dynamics along pseudotime (upper right). Enriched GO terms are also shown (lower right). **l** Heatmap showing the expression levels of selected M44 genes during the trajectories. **m** Boxplot of the M44 expression. Data represent box plots and individual data points. Box plots show the median (center line), first and third quartiles (box edges), while the whiskers going from each quartile to the minimum or maximum. $n = 300$ for C1, $n = 267$ for C2, $n = 196$ for C3. **n** Boxplot of plasma protein levels of IGFBP7 (upper) and NT-proBNP (lower). RFU, relative fluorescence unit. HF, heart failure. HTx, heart transplantation. Data represent box plots and individual data points. Box plots show the median (center line), first and third quartiles (box edges), while the whiskers going from each quartile to the minimum or maximum. $n = 768$ for Control, $n = 85$ for non-advanced HF, $n = 30$ for advanced HF before HTx, $n = 30$ for advanced HF after HTx. **o** Mean decrease in accuracy (in order of decreasing accuracy from top to bottom) of proteins involved in classification between manifest and advanced heart failure as assigned by the random forest classifier. **p** Receiver-operating characteristic (ROC) curves based on plasma protein levels of NT-proBNP and IGFBP7 for predicting the incidence of heart failure. Area under the curve is also shown for each marker. NT-proBNP had significantly better ability to diagnose heart failure ($P$-value = 1.109e-07). Areas under the ROC curve were calculated using logistic regression. **q** ROC curves based on plasma protein levels of NT-proBNP and IGFBP7 for discriminating patients with advanced heart failure. Area under the curve is also shown for each marker. Although not statistically significant, IGFBP7 had a higher power to determine the severity of heart failure ($P$-value = 0.1063). Areas under the ROC curve were calculated using logistic regression.

microscope (Keyence). The following primary antibodies were used for fluorescent immunostaining: rabbit monoclonal anti-pSmad2/3 (Ser456/467) antibody (#8828, Cell Signaling Technology, 1:200), rabbit monoclonal anti-p21 antibody (ab188224, Abcam, 1:200), mouse monoclonal anti-γH2A.X (Ser140) antibody (#MA1-2022, Thermo Fisher Scientific, 1:200), rabbit monoclonal anti-PDGFRa antibody (#3174, Cell Signaling Technology, 1:200).

To quantify the γH2A.X positive nuclei in the heart, we obtained the images of γH2A.X staining from both WT and Htra3 KO mice after TAC operation ($n = 3$ each). Four images were taken from each heart. Raw imaging data were analyzed using BZ-X analyzer software (Keyence Corp.) to quantify the fluorescence intensity of the γH2A.X signal merged with DAPI in each nucleus. Subsequently, we set the threshold to detect γH2A.X-positive nuclei and the software automatically calculated the percentage of γH2A.X-positive nuclei. All raw imaging data were analyzed using the same algorithm[14].

To measure the amount of fibrosis, sections were stained with Picrosirius Red/ Fast Green dyes. After captioning the whole image of sections, the percentage of LV fibrotic area was assessed by areas of red- and green-stained regions, using BZ Analyzer software (Keyence). All histological quantifications were performed by 2 observers in a blinded manner.

**AAV infection.** The AAV vectors were prepared by VectorBuilder Inc. (https://en.vectorbuilder.com) according to established procedures[73]. Briefly, AAV vectors of serotypes 2 and 9 were generated in HEK293T cells, using triple-plasmid co-transfection for packaging. Viral stocks were obtained by CsCl2-gradient centrifugation. Titration of AAV viral particles was performed by real-time PCR quantification of the number of viral genomes, measured as CMV copy number. The viral preparations had a titer between $1 \times 10^{12}$ and $5 \times 10^{12}$ genome copy (GC)/mL. Viruses were administered in 100 μL saline via tail-vein injection. For experiments performed in the HtrA3 KO mice, $3 \times 10^{11}$ GC doses of rAAV9-GFP or $3 \times 10^{11}$ GC doses of rAAV9-shNox4 were administered to the uninjured mice 1 week before TAC surgery. For experiments of Htra3 overexpression, $3 \times 10^{11}$ GC doses of rAAV9-GFP or rAAV9-Htra3 were administered into WT mice 1 week after TAC surgery. For experiments of Nox4 overexpression, $3 \times 10^{11}$ GC doses of rAAV9-GFP or rAAV9-Nox4 were administered into WT mice of 8 weeks old.

**qRT-PCR analysis.** For mechanical stretch experiment, cardiac fibroblasts isolated from neonatal mice were plated onto 100 mm quadrangular, gelatin (0.1%) coated silicone rubber culture plates and cultured with serum-free DMEM for 4 h as previously described[74,75]. After 4 h, medium was changed to 2% FBS/DMEM and cultured for another 24 h. Mechanical stretch was introduced to the attached fibroblasts by 10–30% persistent elongation. At 1 day after the stretch, cells were lysed using Smart-seq2 lysis buffer of and then used for cDNA library construction with Smart-seq2. mRNA expression was evaluated by qRT-PCR using a CFX96 Real-Time PCR Detection System, and the relative expression levels of the target genes were normalized to the expression of an internal control gene, using the comparative Ct method. The following primer sets were used for qRT-PCR.

*Rps18* mRNA forward, CTTAGAGGGACAAGTGGCG
*Rps18* mRNA reverse, ACGCTGAGCCAGTCAGTGTA

*Htra3* mRNA forward, TGACCAGTCCGCGGTACAAG
*Htra3* mRNA reverse, TTGGAGCTGGAGACCACGTG
*Tgfb1* mRNA forward, CTCCCGTGGCTTCTAGTGC
*Tgfb1* mRNA reverse, GCCTTAGTTTGGACAGGATCTG

For qRT-PCR of *Htra3*, multiple organs such as heart, placenta, skeletal muscle, brain, liver, kidney, and lung were collected from both *Htra3* KO mice and WT controls after systemic perfusion of cold PBS. Total RNA was isolated from these tissues using TRIzol reagent (#15596026, Thermo Fisher Scientific). After its purity was confirmed by 260/280 nm absorbance (>1.8), single-stranded cDNA was synthesized using the High-Capacity cDNA Reverse Transcription Kit (#4374966, Thermo Fisher Scientific) from 1 μg of RNA following the manufacturer's instructions. mRNA expression was evaluated by qRT-PCR using a CFX96 Real-Time PCR Detection System, and the relative expression levels of the target genes were normalized to the expression of an internal control gene, using the comparative Ct method.

**Plasma proteome analysis.** Plasma proteomic data from different stages of heart failure ($n = 768$ for control subjects, $n = 84$ for non-advanced heart failure, $n = 30$ for advanced heart failure before transplantation, $n = 30$ for advanced heart failure after transplantation) were obtained from Egerstedt et al. and used for proteome analysis[34]. To assess the accuracy of the classification and identify factors for discriminating between groups, the "randomForest" package in R was used. Receiver-operating characteristic analysis was performed to assess diagnostic and discriminating ability, using the "pROC" package in R.

**Statistics and reproducibility.** All statistical tests and graphical depictions of data (means and error bars) are defined within the figure legends for the respective data panels. For comparisons between two groups, unpaired or Student's two-tailed $t$-tests were performed as noted within the figure legends. For comparisons between more than two groups, one-way or two-way analysis of variance was performed as noted within the figure legends. Unless otherwise indicated, $P < 0.05$ was considered as statistically significant. In the boxplots, horizontal lines indicate the medians, boxes show the 25th–75th percentile, and whiskers represent the minimum and maximum values. Imaging experiments (Figs. 1j, k, 4j, 5c and Supplementary Figs. 3f, 4f, g, 5c, 7f) were performed twice by independent researcher. Western blotting experiments (Figs. 3k, l, 5d, e, g and Supplementary Figs. 1k, 3e) were performed once using protein from different samples. The in vitro experiments such as imaging studies and western blotting were conducted and checked together by at least 2 independent researchers.

**Reporting summary.** Further information on research design is available in the Nature Research Reporting Summary linked to this article.

## Data availability

Source data are provided with this paper. The source data underlying Supplementary Fig. 1j, Figs. 2b–d, g, 3m, Supplementary Fig. 3g, h, Supplementary Fig. 4h, Fig. 5b, h, Supplementary Fig. 5b and Fig. 6c are provided as a Source Data file. The RNA-seq data for this study has been deposited in the Gene Expression Omnibus under accession

number GSE168742. Single-cell RNA-seq data of cardiomyocytes from p53 knockout mice have been deposited under accession number GSE 95143. The information of mouse genome (mm9) is available on UCSC Genome Browser (http://genome.ucsc.edu). Any additional data supporting the findings of this study other than deposited data described previously are available from the authors on reasonable request.

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

## Acknowledgements

We thank K. Shiina for next-generation sequencing support, Y. Yokota and N. Furukawa for experimental support, and H. Nakajima for proofreading. This work was supported by grants from a Grant-in-Aid for Young Scientists (to T.K.), the Japan Foundation for Applied Enzymology (to S.N. and T.K.), the SENSHIN Medical Research Foundation (to S.N. and T.K.), the Kanae Foundation for the Promotion of Medical Science (to S.N.), MSD Life Science Foundation (to S.N. and T.K.), The Tokyo Biomedical Research Foundation (to S.N.), Astellas Foundation for Research on Metabolic Disorders (to S.N.), The Novartis Foundation (Japan) for the Promotion of Science (to S.N.), the Japanese Circulation Society (to S.N.), the Takeda Science Foundation (to S.N.), the Uehara Memorial Foundation (to S.N.), a Grant-in-Aid for Scientific Research (B) (to S.N.), a Grant-in-Aid for Scientific Research (A) (to S.N.), a Grant-in-Aid for Scientific Research (S) (to I.K.), AMED (JP22ek0210152, JP21gm6210010, JP20bm0704026, JP22ek0210141, JP22ek0109440, JP22ek0109487, JP22gm0810013, JP22km0405209, JP21ek0210118, JP21ek0109406, JP22ek0109543, JP22ek0109569, JP21tm0724601, JP22ama121016, JP22ek0210172, JP22ek0210167) (to S.N., H.A., and I.K.), JST FOREST Program, Grant Number 21466223 (to S.N.), and a grant from the Cell Science Research Foundation (to S.N.).

## Author contributions

T.K., S.N., H. Aburatani, and I.K. conceived the project, designed the study, and interpreted the results; T.K., S.N., and K.F. collected single cells and generated the single-cell sequencing data; S.N. and S.Y. performed computational analyses; T.F. and H. Aburatani provided support for computational analyses; T.K. performed the TAC and MI procedures, conducted functional analysis, performed biochemical experiments, and analyzed the data; C.O. generated *Htra3* KO mice; M.S., M. Katoh, S.Y., M. Ito, M. Katagiri, T.S., B.Z., S.H., T.Y., M. Harada, H.T., E.A., M. Hatano, O.K., K.N., H. Abe, T.U., M.O., M. Ikeuchi, H.M., H. Aburatani, and I.K. provided experimental and analytical support; T.K., S.N., H. Aburatani, and I.K. wrote the manuscript with input from all authors.

## Competing interests

The authors declare no competing interests.
