## [Peer Review File · Nature Communications]

nature portfolio

Peer Review File

Draft OnlyREVIEWER COMMENTS

Reviewer #1 (Remarks to the Author):

The study examines the role of the serine peptidase Htra3 in cardiac homeostasis and in heart failure. The authors show that Htra3 is predominantly expressed in cardiac fibroblasts and that global germline loss of Htra3 causes a hypertrophic cardiomyopathy that is further accentuated in response to pressure overload induced through TAC. The effects of Htra3 are mediated through TGF-beta degradation: suppression of endogenous Htra3 following pressure overload induces an exaggerated TGF-beta response.

General comment:

The study deals with an interesting and novel concept. The data are generally of high quality. The main problems relate to the exclusive reliance on a global KO line (and the absence of fibroblast-specific approaches), and the overinterpretation of the findings to support an effect of fibroblasts on cardiomyocyte senescence. The following concerns need to be addressed:

Major comments:

1. The main weakness of the study is the use of a global Htra3 KO line to suggest fibroblast-specific effects. The authors attempt to circumvent this by showing single cell data suggesting that Htra3 is predominantly expressed in fibroblasts. However, considering that the protein is expressed at high levels in the myocardium and in other muscle tissues (compared to other organs), expression at the protein level in cardiomyocytes (and in other muscle cells) is likely and may be involved in regulation of cell size. Conclusions regarding the fibroblast-specific effects of Htra3 require fibroblast-specific loss-of-function approaches. The authors should document the predominant expression of Htra3 in fibroblasts vs cardiomyocytes at the protein level, and acknowledge the limitation related to the lack of cell-specific loss-of-function approaches.

2. Htra3 KO mice need to be characterized. Mice with global loss of Htra3 had significant baseline hypertrophy. What is the basis for this abnormality? Was systemic blood pressure affected? Were there other effects on cardiomyocytes? Were there any effects in other organs? If Htra3 is expressed in all fibroblasts and inhibits TGF-beta signaling, abnormalities in other organs may be prominent.

3. The cardiomyopathic response in Htra3 KOs needs to be characterized. Is this progressive? Does it result in heart failure and mortality in the absence of injury?

4. The authors suggest that the phenotype in Htra3KOs is driven by overactive TGF-beta. Does TGF-beta neutralization abolish the hypertrophic response at baseline?

5. How do the authors explain the absence of baseline fibrosis in mice lacking Htra3?

6. The criteria used by the authors to suggest cardiomyocyte “senescence” are problematic. They seem to define senescence as practically any activation of oxidative responses, or induction of certain inflammatory genes. DNA damage is not documented. The authors need to tone down statements regarding cell senescence and simply refer to their observations, limiting the use of speculative statements.

7. There is a tendency to overinterpret. The title epitomizes this problem: “Cardiac fibroblasts are critically involved in the development of heart failure by inducing cardiomyocyte senescence”. However, fibroblast-specific approaches are lacking and cardiomyocyte senescence is not demonstrated (unless one uses the term senescence to describe any injury). Please revise the title and conclusions, interpreting the findings.

8. Data on the fibroblast-specific expression of Htra3 need to be strengthened. Please provide comparative analysis of levels in different myocardial cell types. Was Htra3 expressed in all fibroblast subsets? The images documenting Htra3 expression in Pdgfra+ cells need to be improved. It is impossible to appreciate the colocalization.

Minor:

Htra3 was identified as a molecule located “at the center of the network”. Please explain what that means.

Results: “To investigate molecular interactions leading to the induction of senescent failing cardiomyocytes”. Why did the authors assume that cardiomyocytes become “senescent” following pressure overload? The study investigates effects of pressure overload not cell senescence.

The abstract requires minor revisions and clarifications: a. please explain the rationale for the focus on Htra3, b. “governing the identity”, perhaps a more specific term would be preferable.

Reviewer #2 (Remarks to the Author):

By using single-cell RNA-seq, spatial transcriptomics, and genetic modifications the authors showed that high-temperature requirement A serine peptidase 3 (Htra3) is a critical regulator of cardiac fibrosis and cardiomyocyte senescence. Htra3, specifically expressed in cardiac fibroblasts keeps them quiescent and prevents TGF- β degradation. TGF- β is known inducer of so called secondary senescence, but the authors showed that Htra3-TGF- β -IGFBP7 pathway is responsible for cardiac

fibroblast activation and cardiomyocyte senescence. They provided evidence that cardiomyocyte senescence is DNA damage dependent

governing the identity of quiescent cardiac fibroblasts through degrading TGF- β .

I have some comments concerning this issue:

1. It cannot be excluded that in KO Htra3 deficient mice after intense proliferation and collagen production also cardiac fibroblasts undergo senescence and in this state non-degraded TGF- β induces secondary senescence in cardiomyocytes. Moreover SASP of senescent fibroblasts can be fibrogenic (doi.org/10.1038/ncomms14532). Can the authors exclude senescence of fibroblasts in their model of heart failure?

2. DNA damage-dependent senescence of cardiomyocytes could be better documented (Fig.4). Double positive (γ H2AX, p21) nuclei are hardly visible. What about quantitative analysis? The levels of p53 and/or ATM would be desirable.

3. Both Nox4 overexpression (Lener et al., 2009) and downregulation can induce cell senescence. Moreover it can be DNA damage-independent (Przybylska et al, 2016). Thus, it is necessary to document the senescence state by showing SA- β -gal activation upon Nox4 overexpression and generally in any cells supposed to be senescent.

Reviewer #3 (Remarks to the Author):

Cardiac fibroblasts are critically involved in the development

of heart failure by inducing cardiomyocyte senescence by Toshiyuki et al is an immense and outstanding work.

By genetic perturbation in several models and with scRNAseq and spatial transcriptomics the authors identify fibroblast expressed Htra3 as a critical regulator and its downregulation via activated TGF- β signalling of cardiac fibrosis and cardiomyocyte senescence through DNA damage. The Htra3-TGF- β -IGFBP7 was explored in mouse pressure overload, AMI and knockout models. Further, it was confirmed in human failing heart and demonstrated that the cytokine IGFBP7 is secreted from senescent cardiomyocytes and is a putative therapeutic target in heart failure.

Seminal for scientific results is detailed description of methods so they and results can be critically scrutinized. Here is a list on what might be improved.

1. Ethics: How was heart samples for human controls obtained. What was the time delay between death and tissue isolation? How does it compare to patient samples? It is not trivial how live human cardiomyocytes can be obtained from a healthy donor.
2. Line 501: How were live human cardiomyocytes isolated and processed? It is not described in ref 1. In Ext data fig 2b UMAP of human cells are shown for a number of cell types. Where these isolated and processed as for the human cardiomyocytes?
3. Mouse cardiomyocytes were isolated by Langendorff perfusion. To what extent are they representative for the whole heart? What was the yield? Did you consider using snRNAseq to get RNA for all cell types in the same analysis?
4. Cluster analysis: It seems that both WGCNA, Seurat 3 and Seurat 4 were used for cluster analysis. Why did you use different methods? What were the conditions used in the different analyses? What was the results of quality analysis? When you merged clusters analysed for different samples, did you correct for batch effects?
5. Spatial transcriptomics: Which method did you use for the projection of scRNAseq clusters on spatial transcriptomics maps? Which scRNAseq cell type classification have been used? Is it from fig 1 where classification was done with WGCNA, another method of classification. If so, has this procedure been validated? Or is it de novo scRNAseq from cells in the infarct zone as indicated in the text? In that case, the methods, results and analysis of the scRNA is not given in results and methods sections.
6. Fig 6: You show spatial transcriptomics w 5 clusters. Cl3 seem to be mainly characteristics of blood. You show projections of scRNAseq cell types. How was this done? What does the proportions mean? For example does a proportion of 0.4 mean that 40% of the cells were of that type? If so, the number of cells in cl2 is much more than possible 100%.
7. Number of detected genes: In the different experiments you used different cutoff for the genes to be analysed. What principle did you use to decide on the cutoff?
8. Figure 1: g: Coexpression network of cardiac fibroblasts. As I understand this is the basic analysis to decide on exploration of Htra-3. Why did you decide on only HTRA-3 and not the other main genes in the figure? h: Top genes correlated with the fibroblast module. This method is only superficially described in the methods section and there is no xls file on the genes related to this module or any other.
9. The only two xls files provided are computed tables and the only on basic analysis ligand-receptor pairs between modules. This table is perplexing:

For example NPPA: Ligand module 0, receptor module 2. Where 0 may be interpreted as no cell type and 2 as M-2 endothelial cell.

What does zero mean? NPPA have no ligand module? Npr2 no receptor module?
10. It is difficult to get an overview of the 44 modules and what they stand for. How were they discriminated and what genes were systematically identified?

11. Line 309 It is stated that IGFBP7 had higher diagnostic power than NT-proBNP. The statistical difference between the ROC curves for this conclusion is not given.

12. Line 213: where are the scRNAseq data in fig 5b?

Draft Only

Responses to the reviewers' comments

We thank all reviewers for their insightful comments regarding our manuscript. In view of the reviewers' suggestions, we have performed additional analyses and revised the manuscript. The sentences that were revised according to the reviewers' comments are highlighted in yellow in the main text.

Reviewer #1 (Remarks to the Author):

Major comments:

1. *The main weakness of the study is the use of a global Htra3 KO line to suggest fibroblast-specific effects. The authors attempt to circumvent this by showing single cell data suggesting that Htra3 is predominantly expressed in fibroblasts. However, considering that the protein is expressed at high levels in the myocardium and in other muscle tissues (compared to other organs), expression at the protein level in cardiomyocytes (and in other muscle cells) is likely and may be involved in regulation of cell size. Conclusions regarding the fibroblast-specific effects of Htra3 require fibroblast-specific loss-of-function approaches. The authors should document the predominant expression of Htra3 in fibroblasts vs cardiomyocytes at the protein level, and acknowledge the limitation related to the lack of cell-specific loss-of-function approaches.*

Response

We conducted qRT-PCR to verify the specific expression of *Htra3* in the heart among major organs (**Extended Data Fig. 1j**), which was consistent with the previous report (ref. 27). Next, scRNA-seq analysis revealed the expression of *Htra3* was only seen in cardiac fibroblasts among various cell-types in the heart (**Fig. 1i, Extended Data Fig. 1b, Extended Data Fig. 6f**). Finally, we further conducted western blotting using cardiomyocytes and non-cardiomyocytes (including cardiac fibroblasts) isolated with Langendorff perfusion method, showing that *Htra3* protein was predominantly observed in non-cardiomyocytes (**Extended Data Fig. 1k**). Based on these results, *Htra3* is considered to be predominantly expressed in cardiac fibroblasts. We also mentioned the limitation of the lack of cell-specific loss-of-function approaches in the Discussion section in our revised manuscript.

2. *Htra3* KO mice need to be characterized. Mice with global loss of *Htra3* had significant baseline hypertrophy. What is the basis for this abnormality? Was systemic blood pressure affected? Were there other effects on cardiomyocytes? Where there any effects in other organs? If *Htra3* is expressed in all fibroblasts and inhibits TGF- β signaling, abnormalities in other organs may be prominent.

Response

We checked the blood pressure, but there was no significant difference in blood pressure between WT and *Htra3* KO mice (Extended Data Fig. 2b). For organs other than the heart, we could not find any difference in their size and tissue histology (HE staining) (Extended Data Fig. 2d). As we mentioned before, the expression of *Htra3* is predominantly seen in cardiac fibroblasts, which may explain the normal phenotype of other organs. We also compared transcriptomic profiles from WT and *Htra3* KO mice after sham operation. In cardiomyocytes from *Htra3* KO mice, genes involved in oxidative-reduction process were down-regulated and genes involved in protein synthesis were up-regulated without pressure overload (Extended Data Fig. 4e), which may lead to cardiomyocyte hypertrophy in *Htra3* KO mice (Fig. 2c).

3. *The cardiomyopathic response in Htra3* KOs needs to be characterized. Is this progressive? Does it result in heart failure and mortality in the absence of injury?

Response

We compared the natural history of cardiac function between WT and *Htra3* KO mice. As shown in Extended Data Fig. 2c, we could not observe any differences in LVDd/Ds and FS between WT and *Htra3* KO mice with aging up to 60 weeks old.

4. *The authors suggest that the phenotype in Htra3* KOs is driven by overactive TGF- β . Does TGF- β neutralization abolish the hypertrophic response at baseline?

Response

We performed TGF- β neutralization in *Htra3* KO mice without any additional surgery. Since ventricular hypertrophy has already been observed in *Htra3* KO mice of 8 weeks old, we injected anti-TGF- β antibodies from 4 weeks old, showing that TGF- β neutralization successfully inhibited the ventricular hypertrophy in *Htra3* KO mice (Extended Data Fig. 3h).

5. How do the authors explain the absence of baseline fibrosis in mice lacking *Htra3*?

Response

As shown in **Fig. 2d**, we quantified the percentage of fibrosis area. *Htra3* KO mice showed significantly more interstitial fibrosis than WT mice even without TAC surgery.

6. The criteria used by the authors to suggest cardiomyocyte “senescence” are problematic. They seem to define senescence as practically any activation of oxidative responses, or induction of certain inflammatory genes. DNA damage is not documented. The authors need to tone down statements regarding cell senescence and simply refer to their observations, limiting the use of speculative statements.

Response

We agree with the reviewer’s comment. The sentences and words revised according the suggestion were highlighted in yellow in the revised manuscript.

7. There is a tendency to overinterpret. The title epitomizes this problem: “Cardiac fibroblasts are critically involved in the development of heart failure by inducing cardiomyocyte senescence”. However, fibroblast-specific approaches are lacking and cardiomyocyte senescence is not demonstrated (unless one uses the term senescence to describe any injury). Please revise the title and conclusions, interpreting the findings.

Response

We again agree with the reviewer’s comment. According to the suggestion, we revised the title, results, and conclusions, based on the experimental findings.

8. Data on the fibroblast-specific expression of *Htra3* need to be strengthened. Please provide comparative analysis of levels in different myocardial cell types. Was *Htra3* expressed in all fibroblast subsets? The images documenting *Htra3* expression in *Pdgfra+* cells need to be improved. It is impossible to appreciate the colocalization.

Response

We showed the fibroblast-specific expression of *Htra3* in the heart using violin plot (**Extended Data Fig. 1b**). After myocardial infarction, the fibroblast population was divided into 4 clusters (**Extended Data Fig. 6c**). Fibroblast clusters 2 (FB2) and 4 (FB4), which were increased in the early phase after myocardial infarction, showed lower expression levels of *Htra3* (**Extended Data Fig. 6d-f**), which is consistent with the

finding that expression of *Htra3* in cardiac fibroblasts was down-regulated after pressure overload to the heart (Fig. 2e) or mechanical stretch (Fig. 2g).

We sincerely apologize for the confusing description regarding the images documenting *Htra3* expression in the original paper. Since it was difficult to perform immunostaining and *in situ* hybridization in the same tissue slide, we used paired mirror cardiac sections to perform immunostaining of *Pdgr- α* and *in situ* hybridization of *Htra3*. Therefore, same cells would appear in both sections. In Fig. 1j, Arrows indicate the colocalization of *Pdgr- α* (immunostaining) and *Htra3* (*in situ* hybridization) in the same cells appeared in both sections.

Minor:

Htra3 was identified as a molecule located “at the center of the network”. Please explain what that means.

Response

Since we conducted weighted gene coexpression network analysis to derive the fibroblast network, the central location of *Htra3* in Fig. 1g means that its expression was strongly correlated with the expression of other cardiac fibroblast module genes. By calculating the correlation coefficient of each gene expression with the fibroblast module expression, we also found that *Htra3* expression was strongly correlated with expression of the cardiac fibroblast module (Fig. 1h), suggesting that *Htra3* defines the identity of cardiac fibroblasts.

Results: “To investigate molecular interactions leading to the induction of senescent failing cardiomyocytes”. Why did the authors assume that cardiomyocytes become “senescent” following pressure overload? The study investigates effects of pressure overload not cell senescence.

Response

As the reviewer states, we investigated the effects of pressure overload and revealed the appearance of failing cardiomyocytes with DNA damage and secretory phenotype. We revised the manuscript according to the suggestions.

The abstract requires minor revisions and clarifications: a. please explain the rationale for the focus on Htra3, “governing the identity”, perhaps a more specific term would be preferable.

Response

Thank you for your suggestion. We conducted single-cell RNA-seq of cardiac fibroblasts isolated from WT and *Htra3* KO mice to show that *Htra3* basically inhibits TGF- β signalling and that pressure overload and *Htra3* deletion synergistically activates TGF- β signalling, leading to activation of fibroblasts. Therefore, we changed the sentence from “governing the identity of quiescent cardiac fibroblasts” to “maintaining the identity of quiescent cardiac fibroblasts”.

Reviewer #2 (Remarks to the Author):

1. It cannot be excluded that in KO Htra3 deficient mice after intense proliferation and collagen production also cardiac fibroblasts undergo senescence and in this state non-degraded TGF- β induces secondary senescence in cardiomyocytes. Moreover, SASP of senescent fibroblasts can be fibrogenic (doi.org/10.1038/ncomms14532). Can the authors exclude senescence of fibroblasts in their model of heart failure?

Response

We performed immunostaining of γ H2A.X and Pdgfr-a to show that accumulation of DNA damage was observed not only in cardiomyocytes but also in cardiac fibroblasts in *Htra3* KO mice after TAC surgery (**Extended Data Fig. 4g, h**). DNA damage in cardiac fibroblasts may also induce fibroblast senescence, consistent with our single-cell RNA-seq finding that cardiac fibroblasts of *Htra3* KO mice expressed genes involved in senescence-associated secretory phenotype (e.g., *Igfbp7* and *Tgfb3*) (**Fig. 3h, i, j**).

2. DNA damage-dependent senescence of cardiomyocytes could be better documented (Fig.4). Double positive (γ H2AX, p21) nuclei are hardly visible. What about quantitative analysis? The levels of p53 and/or ATM would be desirable.

Response

We sincerely apologize for the confusing presentation in the **Fig. 4j**. Since it was difficult to identify double positive nuclei of γ H2A.X and p21 in the merged figure, we showed single staining figure separately in **Extended Data Fig. 4f**. We also performed quantitative analysis of γ H2A.X positive cells by calculating the percentage of γ H2A.X

positive nuclei (shown in Extended Data Fig. 4h). We further showed that expression of DNA damage-related genes, such as *Cdkn1a* and *Trp53*, was up-regulated in cardiomyocytes of *Htra3* KO mice after TAC surgery (Fig. 4k).

3. *Both Nox4 overexpression (Lener et al., 2009) and downregulation can induce cell senescence. Moreover it can be DNA damage-independent (Przybylska et al, 2016). Thus, it is necessary to document the senescence state by showing SA-β-gal activation upon Nox4 overexpression and generally in any cells supposed to be senescent.*

Response

We injected the AAV9-Nox4 overexpression vector in WT or *Htra3* KO mice and showed that Nox4 overexpression in the heart increased DNA damage and reduced cardiac function (Extended Data Fig. 5b, c), which was consistent with the previous report (*Proc Natl Acad Sci USA*. 2010; 107(35): 15565-70). Although we revealed that failing cardiomyocytes showed DNA damage accumulation and secretory phenotype, SA-β-gal staining did not work well in *in vivo* samples. Therefore, we changed the sentence from “senescent cardiomyocytes” to “failing cardiomyocytes with DNA damage and secretory phenotype” throughout the manuscript.

Reviewer #3 (Remarks to the Author):

1. *Ethics: How was heart samples for human controls obtained. What was the time delay between death and tissue isolation? How does it compare to patient samples? It is not trivial how live human cardiomyocytes can be obtained from a healthy donor.*

Response

For healthy controls who died due to non-cardiac causes, we obtained heart tissue during the autopsy which took place within 1 hour after declaration of death. On the other hand, for patients with heart failure, we obtained heart tissue within 30 minutes after the tissue was cut from the heart during surgery. Therefore, in single-cardiomyocyte RNA-seq of human samples, the difference in the time from death (for control subjects) or tissue sampling (for patients with heart failure) to cell isolation may generate some bias in transcriptomes. However, we set the same cutoff (detected genes >2,000) for all scRNA-seq data for subsequent analysis in quality control and detected gene modules corresponded to gene modules detected from murine single-cardiomyocyte RNA-seq. We also identified some secretory factors which are

secreted from failing cardiomyocytes and showed they are associated with the severity of heart failure by integrating with plasma proteome analysis. We mentioned this limitation in the Discussion section.

2. Line 501: How were live human cardiomyocytes isolated and processed? It is not described in ref 1. In Ext data fig 2e UMAP of human cells are shown for a number of cell types. Where these isolated and processed as for the human cardiomyocytes?

Response

We apologize for the insufficient description in methodology. Immediately after the collection of the heart tissue, tissue was minced and incubated in lysis buffer containing 2 mg/mL type 2 collagenase (Worthington), 1 mg/mL dispase (Roche), and 20 U/mL DNase I (Roche). After 4 cycles of lytic digestion by mild shaking for a total 20 min at 37 °C, rod-shaped live cardiomyocytes were isolated. We added this description in the Method section. The UMAP plots in Extended Data Fig. 2e, f were generated using the data from reference #4 (Wang, L. *et al.* Single-cell reconstruction of the adult human heart during heart failure and recovery reveals the cellular landscape underlying cardiac function. *Nat. Cell Biol.* **22**, 108–119 (2020)). In ref.4, Wang L et al. used iCell8 (TaKaRa Bio USA) to perform scRNA-seq.

3. Mouse cardiomyocytes were isolated by Langendorf perfusion. To what extent are they representative for the whole heart? What was the yield? Did you consider using snRNAseq to get RNA for all cell types in the same analysis?

Response

Although there is no valid data about the number of cells in the murine whole heart, we isolated more than 10^5 cells from one mouse heart, which was equivalent to the previous reports (such as *Nature* 497, 249–253 (2013) and *Nature* 582, 271–276 (2020)). Besides, we also confirmed that there were no clumps of cardiac tissue after isolation of cardiomyocytes through Langendorf perfusion. Actually, we performed single-nucleus RNA-seq of the heart, but did not quantitatively detect many genes; therefore, we obtained single-cell expression profiles of cardiomyocytes by the Smart-seq2 full-length method and non-cardiomyocytes by droplet-based Chromium Controller (10x Genomics).

4. *Cluster analysis: It seems that both WGCNA, Seurat 3 and Seurat 4 were used for cluster analysis. Why did you use different methods? What were the conditions used in the different analyses? What were the results of quality analysis? When you merged clusters analysed for different samples, did you correct for batch effects?*

Response

Since single-cell RNA-seq through full-length cDNA synthesis by Smart-seq2 quantitatively detected many genes, we applied weighted gene co-expression network analysis to detect gene modules and use their expression levels for clustering analysis. The other single-cell RNA-seq data were analyzed using Seurat with the 'FindIntegrationAnchors' function to correct batch effects. Seurat V4 was applied only for spatial transcriptomics data analysis. We added these explanations in the Methods section.

5. *Spatial transcriptomics: Which method did you use for the projection of scRNAseq clusters on spatial transcriptomics maps? Which scRNAseq cell type classification have been used? Is it from fig 1 where classification was done with WGCNA, another method of classification? If so, has this procedure been validated? Or is it de novo scRNAseq from cells in the infarct zone as indicated in the text? In that case, the methods, results and analysis of the scRNA is not given in results and methods sections.*

Response

We apologize for the sufficient description in methodology. To predict the proportion of cell types in each spot, predict.score was calculated by using 'FindTransferAnchors' and 'TransferData' functions in Seurat (**Fig. 6f**). Dot size represents the average of predict.score of each spot in each cluster. The scRNA-seq data of cells isolated from mice after sham or MI operation (**Extended Data Fig. 6**) were used as reference data. We added these descriptions in the Methods section.

6. *Fig 6: You show spatial transcriptomics w 5 clusters. Cl3 seem to be mainly characteristics of blood. You show projections of scRNAseq cell types. How was this done? What does the proportions mean? For example does a proportion of 0.4 mean that 40% of the cells were of that type? If so, the number of cells in cl2 is much more than possible 100%.*

Response

As the reviewer pointed out, we also consider that since heart sections used for spatial transcriptomics analysis contained red blood cells, cluster 3, which was characteristic for genes related with red blood cells, was generated. However, since we carefully performed perfusion by PBS before and during cell isolation for single-cell RNA-seq, we did not detect red blood cells in our single-cell RNA-seq analysis. As we answered to the previous comment, we used scRNA-seq data of cells isolated from mice after sham or MI operation as the reference, and re-analyzed to predict cell types in each spot and derive the proportion of cell types in each cluster (Fig. 6f).

7. Number of detected genes: In the different experiments you used different cutoff for the genes to be analysed. What principle did you use to decide on the cutoff?

Response

In single-cell RNA-seq through full-length cDNA library synthesis by Smart-seq2, we calculated detected genes for each cell and generated histogram to set the cutoffs for genes to be analyzed (Extended Data Fig. 1c, 3a, 4a, and 7a). We added this explanation in the Methods section.

8. Figure 1: g: Coexpression network of cardiac fibroblasts. As I understand this is the basic analysis to decide on exploration of Htra-3. Why did you decide on only HTRA-3 and not the other main genes in the figure? h: Top genes correlated with the fibroblast module. This method is only superficially described in the methods section and there is no xls file on the genes related to this module or any other.

Response

We calculated the correlation coefficient between each gene expression with fibroblast module expression in Fig. 1h. We also generated Supplementary Table 3 as original matrix data for Fig. 1h. We added these explanations in the Methods section. In Fig. 1h, we detected genes significantly correlated with fibroblast module expression other than *Htra3* (e.g., *Dcn* and *Islr*), but these genes have already been shown to be involved in the pathogenesis of cardiac disease via TGF- β signaling (ref. 24 and 25). Therefore, we focused on *Htra3* in this study.

9. The only two xls files provided are computed tables and the only on basic analysis ligand-receptor pairs between modules. This table is perplexing:

For example NPPA: Ligand module 0, receptor module 2. Where 0 may be interpreted as no cell type and 2 as M-2 endothelial cell.

What does zero mean? NPPA have no ligand module? Npr2 no receptor module?

Response

We apologize for lacking the detailed explanation about **Supplementary Table 1**. As the reviewer predicted, zero means that there is no corresponding cell type. We added this explanation in the revised version of **Supplementary Table 1**. For example, in the line showing the interaction between Nppa (ligand) and Nrp1 (receptor), Nppa (ligand) is not expressed in specific cell types (M0), whereas Nrp1 (receptor) is specifically expressed in endothelial cells (M2). On the other hand, in the line showing the interaction between Nppb (ligand) and Nrp1 (receptor), Nppb (ligand) is specifically expressed in cardiomyocytes (M4), whereas Nrp1 (receptor) is specifically expressed in endothelial cells (M2).

10. It is difficult to get an overview of the 44 modules and what they stand for. How were they discriminated and what genes were systematically identified?

Response

By calculating the overlap between gene modules detected from single-cardiomyocyte RNA-seq in human and mouse, we identified the statistically significant overlap between M1 (human) and M2 (mouse), between M2 (human) and M1 (mouse), and between M44 (human) and M9 (mouse) (**Extended Data Fig. 7d**). Genes involved in extracellular matrix organization and TGF-beta receptor signaling pathway were enriched in M44 (**Fig. 7k**). Expression dynamics of representative genes in M44 along with pseudotime are shown in **Fig. 7k, l**.

11. Line 309 It is stated that IGFBP7 had higher diagnostic power than NT-proBNP. The statistical difference between the ROC curves for this conclusion is not given.

Response

The statistical difference between the ROC curves in **Fig. 7q** was P-value = 0.1063. We stated this in the Figure legend of **Fig. 7q**.

12. Line 213: where are the scRNAseq data in Ext fig 5b?

Response

We used the data from our previous report (Nomura, S. *et al.* Cardiomyocyte gene programs encoding morphological and functional signatures in cardiac hypertrophy and failure. *Nat. Commun.* **9**, 4435 (2018)) to generate **Extended Data Fig. 5d** (**Extended Data Fig. 5b** in the original version). We added this statement in the Figure legend of **Extended Data Fig. 5d**.

Finally, the authors would like to thank the editor and reviewers again for these valuable comments and suggestions.

Draft Only

REVIEWERS' COMMENTS

Reviewer #1 (Remarks to the Author):

The authors have been responsive and have addressed most of my concerns. I have no further recommendations.

Reviewer #2 (Remarks to the Author):

The Authors adressed my comments properly. I have no more comments.

Reviewer #3 (Remarks to the Author):

The ms has been improved considerably. Still it is unclear from the text how the modules were defined. It should be clear from the Methods text. Far fetched is to find In the xls table Modules 1-7 are defined as a specific cell type. Why then not to call it a cell type instead of a module, or a name containing the cell type? Also module 44 is still not clearly defined in the text. Should that be cells with both ligand and receptor at cardiomyocytes? Methods for PATHWAY, GO and COEXPRESSION NETWORK ANALYSIS are not clearly stated and referenced. Marker genes, for example for Fig 1i are difficult to find with their references. Language can be improved in the sentence at lines 550-553.

Responses to the editor's and the reviewers' comments

We thank the editor and all reviewers for their insightful comments regarding our manuscript. In view of the reviewers' suggestions, we have revised the manuscript. The sentences that were revised according to the reviewers' comments are highlighted in yellow in the main text.

Reviewer #3 (Remarks to the Author):

Comments:

1. *The ms has been improved considerably. Still it is unclear from the text how the modules were defined. It should be clear from the Methods text. Far fetched is to find In the xls table Modules 1-7 are defined as a specific cell type. Why then not to call it a cell type instead of a module, or a name containing the cell type? Also module 44 is still not clearly defined in the text. Should that be cells with both ligand and receptor at cardiomyocytes? Methods for PATHWAY, GO and COEXPRESSION NETWORK ANALYSIS are not clearly stated and referenced. Marker genes, for example for Fig 1i are difficult to find with their references. Language can be improved in the sentence at lines 550-553.*

Response

We deeply appreciate the Reviewer's comments.

For constructing the LR interaction network map in the heart, we used weighted co-expression network analysis to generate co-expression gene modules and annotate cell-type-specific gene modules by using cell-type-specific gene expression profiles, which were obtained by UMAP projection of single-cell data. For weighted co-expression network analysis, all genes expressed at an FPKM value of ≥ 10 in at least one of the samples were used to construct a signed network using the WGCNA R package, which was also used for cell type annotation in Supplementary Data 1 (The ligand and receptor interaction pairs in the heart). The soft power threshold was analyzed with the "pickSoftThreshold" function and applied to construct a signed network and calculate the module eigengene expression using the "blockwiseModules" function. Modules with < 30 genes were merged with their closest larger neighbouring module. To visualize the weighted co-expression networks, Cytoscape (version 3.7.2) with "edge-weighted force-directed" was used. We described these explanations in the Methods section

(page 15 line 29).

In single-cell RNA-seq analysis of human cardiomyocytes, we performed Random forest and overlap analysis to identify co-expression gene modules M1, M2, and M44 as significantly involved in cell classification and conserved between human and mouse (Supplementary Fig. 7b-d). We did not perform ligand receptor interaction network analysis using human cardiomyocytes in the manuscript. By calculating the overlap between gene modules detected from single-cardiomyocyte RNA-seq in human and mice, we found that M44, which corresponded to murine cardiomyocyte M9 (Supplementary Fig. 7d), was enriched with genes involved in extracellular matrix organization and activated specifically in trajectory 2 (Fig. 7k-m). We described these explanations in the main text (page 10 line 5 and page 10 line 19).

We also added the description about Pathway, GO, and co-expression analysis in the Methods section in our revised manuscript (page 16 line 29).

For cell-type classification of single-cell analysis in Fig. 1i (mouse) or Supplementary Fig. 2e (human), we extracted genes characteristic for each cluster in the UMAP plot (for example, *Myl2*, *Myl4*, *Myh6*, and *Myh7* for cardiomyocyte, *Kdr*, *Fabp4*, and *Vwf* for endothelial cell, *Col1a1*, *Dcn*, and *Lum* for fibroblast) and annotate cell-types for each cluster. Marker gene expression profiles are shown in Supplementary Fig. 1a and b (mouse) and Supplementary Fig. 2f (human).

Since we didn't know where line 550-553 in the reviewer's manuscript corresponded to in our manuscript, we revisited the manuscript thoroughly and entirely, including the Methods section. The revised sentences are highlighted in yellow in the manuscript.

Finally, the authors would like to thank the editor and reviewers again for these valuable comments and suggestions.